Quality assuring the quality assurance tool: applying safety-critical concepts to test framework development

Thörn Jonathan jonathan.thorn@westermo.com 1
Strandberg Per Erik 1 2
Sundmark Daniel 2
Afzal Wasif wasif.afzal@mdh.se 2
1 Westermo Network Technologies AB , Västerås , Sweden
2 Mälardalen University , Västerås , Sweden
Wagner Stefan
Electronic publication date: 2022 Oct 28
Publication date: 2022
Volume: 8
Electronic Location ID: e1131
Received 2022 Jan 31; Accepted 2022 Sep 22
Copyright: ©2022 Thörn et al.
Copyright year: 2022
Copyright holder: Thörn et al.
License: This is an open access article distributed under the terms of the Creative Commons Attribution License, which permits unrestricted use, distribution, reproduction and adaptation in any medium and for any purpose provided that it is properly attributed. For attribution, the original author(s), title, publication source (PeerJ Computer Science) and either DOI or URL of the article must be cited.
License URL: https://creativecommons.org/licenses/by/4.0/

Keywords: Quality assurance, Test automation, Agile processes, Safety-critical development, Case study, Hybrid processes

Funding: Westermo Network Technologies AB, the Knowledge Foundation grant 20150277 The European Union’s Horizon 2020 research and innovation program 871319 & 957212 This research was funded by Westermo Network Technologies AB, the Knowledge Foundation grant 20150277 (ITS ESS-H), and the European Union’s Horizon 2020 research and innovation program under grant agreement nos. 871319 & 957212. There was no additional external funding received for this study. The funders had no role in study design, data collection and analysis, decision to publish, or preparation of the manuscript.

==============================
The quality of embedded systems is demonstrated by the performed tests. The quality of such tests is often dependent on the quality of one or more testing tools, especially in automated testing. Test automation is also central to the success of agile development. It is thus critical to ensure the quality of testing tools. This work explores how industries with agile processes can learn from safety-critical system development with regards to the quality assurance of the test framework development. Safety-critical systems typically need adherence to safety standards that often suggests substantial upfront documentation, plans and a long-term perspective on several development aspects. In contrast, agile approaches focus on quick adaptation, evolving software and incremental deliveries. This article identifies several approaches of quality assurance of software development tools in functional safety development and agile development. The extracted approaches are further analyzed and processed into candidate solutions, i.e., principles and practices for the test framework quality assurance applicable in an industrial context. An industrial focus group with experienced practitioners further validated the candidate solutions through moderated group discussions. The two main contributions from this study are: (i) 48 approaches and 25 derived candidate solutions for test framework quality assurance in four categories (development, analysis, run-time measures, and validation and verification) with related insights, e.g., a test framework should be perceived as a tool-chain and not a single tool, (ii) the perceived value of the candidate solutions in industry as collected from the focus group.

Introduction

The quality of embedded systems, both the software solution and the hardware platform, is often demonstrated by the results of performed tests and assured by the quality of the solution used to perform them. Frameworks for software testing1 can also be considered mission-critical, since development decisions rely on the correctness of and confidence in the produced results. Poor test framework quality may lead to the introduction of, or failure to detect errors, as well as unreliable test results that reduces feedback quality, which in turn impedes the development process (Asplund, 2014; Shahin, Babar & Zhu, 2017).

Agile and plan-driven development approaches have historically been seen as each other’s counterparts. Plan-driven approaches are focused on discipline in long term prospects and agile approaches on improvising and using history to adapt to new environments and opportunities. Agile approaches are based on a model where software is evolved and continuously delivered through short iterative cycles. Therefore, the extensive upfront plans, designs and documentation related to plan-driven development are not considered as valuable (Boehm & Turner, 2004; Nerur & Balijepally, 2007).

Similarly, standards for functional safety2 often rely on a plan-driven process with predefined phases, although not necessarily required to be performed in a strictly sequential manner. The production of substantial amounts of documentation and artifacts is used as evidence to argue that the system is acceptably safe. In previous work on combinations of agile and plan-driven methods, the perspective of utilizing agile practices into an already existing plan-driven development process seems to dominate; we have been unable to find previous work on the inverted scenario (see references in ‘Positioning Our Work with Respect to Related Work’). This article targets parts of that gap with respect to quality assurance of software development tools in general, and test framework tools in particular. Under the assumption that adherence to safety standards results in increased dependability, and thus increased confidence in the developed product, lessons learned in the safety-critical domain will be valuable for increasing the quality of development tools in an agile context. In particular, this has value in industrial contexts not compelled by compliance to any safety standard entailing Safety Integrity Level (SIL) classification of products or the tools used to assure them. But instead, utilizing recommendations present in standards or approaches in a practical way to assure test frameworks can be trusted as a guarantee of product quality.

In this article, we explore ways for non-safety related development with agile processes to be inspired by safety-related development to develop reliable frameworks for software testing. Thus, the research assumes that strategies for increased confidence and quality in tools used for automated software testing in non-safety development may be found or created from concepts and strategies related to safety-critical development, while maintaining agile and efficient processes.

This case study started with a literature study to identify how quality assurance of software development tools is performed with regards to functional safety development, as well as applied methodologies in agile or hybrid development philosophies. The extracted approaches and additional knowledge gained were then further processed and analysed into a compiled set of candidate solutions—principles or practices for increased quality of and confidence in an automated software test framework. The candidates were iteratively validated and refined both to suit the industrial context of an intended application and to increase general applicability.

The key findings are: (i) 48 approaches for quality assurance identified from previous work and standards—e.g. to re-develop from scratch while following standards ‘Approaches for Increased Tool Confidence’. The approaches were derived into 22 candidate solutions for test framework quality assurance in the categories of development, analysis, run-time measures, and verification and validation—e.g., to apply measures to avoid development faults introduced by misconceptions (Table 1). (ii) Industrial value of the approaches as perceived by a focus group which also identified an additional three candidate solutions ‘Summary of Approaches from Previous Work’.

Table 1 Candidate solutions for quality assurance of a quality assurance tool (22 from the literature study, and three from the focus group).

The third column links back to main text and the approaches described in Section 3.3. The two rightmost columns describes qualitative and quantitative appraisal from the focus group, explained in Section 4.8.2 –Qual. indicates good idea ‘:-)’, bad idea ‘:-(’ or indifferent opinion ‘:- |’, whereas Quan. shows percentage of effort the focus group would like to invest in the approaches.

Id	Candidate	Based on Ap. #	Qual.	Quan.	
D	Main aspect: Development				
D.1	Apply measures to avoid development faults introduced by misconceptions	9, 22, 24 & 42	:-)	13%	
D.2	Apply restrictions on tool usage	29 & 42	:- |	–	
D.3	Apply measures to avoid potential errors introduced by users	22, 28 & 34	:-)	–	
D.4	Develop the test framework based on requirements	12, 14, 21, 23 & 40	:-(	–	
D.5	Apply measures of rigour to the development process	20 & 25	:- |	–	
D.6	Re-develop the entire test framework with a suitable safety standard	3, 11, 18 & 48	:-(	–	
A	Main aspect: Analysis				
A.1	Perform formal risk and impact analysis	1, 5, 9, 24, 27, 29 & 33	:-)	10%	
A.2	Analyze the tools using a tool error checklist	9	:- |	–	
A.3	Perform analysis with regards to abnormal operating conditions	16, 26, 27 & 45	:-)	–	
A.4	Analyze using well defined peer-reviews during development	15, 17 & 42	:-)	18%	
A.5	Analyze the tools with static analysis	15 & 42	:-)	5%	
A.6	Perform sufficient root-cause analysis on detected errors	–	:-)	–	
R	Main aspect: Run-time measures				
R.1	Develop automated sanity checks of important tool actions	6, 29 & 41	:- |	5%	
R.2	Implement checks of output from a preceding tool in the tool-chain	6, 10 & 41	:- |	–	
R.3	Develop a monitoring system for error detection and prevention	7, 8, 19 & 41	:-)	15%	
R.4	Develop protection against identified abnormal operating conditions	16, 22, 26 & 45	:-)	5%	
R.5	Implement redundancy in tools and tool-chain	29, 36 & 41	:- |	–	
R.6	Halt execution on detection of errors or erroneous conditions	–	:-)	–	
V	Main aspect: Verification & Validation				
V.1	Utilize a suitable safety standard to validate the tool and related processes	3, 31, 37 & 47	:-(	–	
V.2	Formally prove that tool outputs conforms to specification	4	:-)	–	
V.3	Base tool confidence on history of successful use	30, 35, & 46	:- |	–	
V.4	Use a customized tool validation test suite for critical use cases	44	:-)	–	
V.5	Perform tests based on fault injection	2 & 43	:-)	10%	
V.6	Perform unit tests on modules and tools in tool-chain(s)	15	:-)	19%	
V.7	Implement requirement-based testing	–	:-)	–	

Background

Plan-driven development is based in a well-defined, formal, and specific process to achieve a predictable result; great emphasis is placed on layers of traceable requirements, risk management, verification and validation (Hanssen, Stålhane & Myklebust, 2018a). Even before any construction has begun, the properties of the final product are known and can be precisely defined. A number of roles are typically defined and independence between these is often required as a factor of control (Linz, 2014; Hirsch, 2005). Standards for functional safety ‘Industry Standards for Functional Safety’ often assume the use of documentation-heavy plan-driven processes, where the development is often performed as a sequential process through predefined phases (Jonsson, Larsson & Punnekkat, 2012). Standards for functional safety, (e.g., IEC 61508:2010, 2010, ISO 26262:2018, 2018, EN 50128:2011, 2011), often describe the development life cycle as a sequential flow influenced by the V-model (Asplund, 2014), illustrated in Fig. 1.

In contrast to plan-driven development, agile development does not rely on high degrees of documentation or rigid processes. Instead, agile approaches are based on a model where software is evolved and continuously delivered through short iterative cycles with continuous feedback (Linz, 2014). An important aspect of agile is to embrace and respond to changes. Therefore, extensive upfront plans and designs are not considered as valuable. Working software that adds value is prioritized over comprehensive documentation. Important aspects of agile approaches are continuous improvements and code integration, resulting in continuous delivery. Rituals like daily stand-up meetings, demonstrations and reflections provide progress tracking, feedback and process improvements (Nerur & Balijepally, 2007; Dingsøyr et al., 2012). Two popular implementations are Scrum and Kanban (Fowler & Highsmith, 2001). Both use phases which items from the product backlog traverse through, before being packaged for release, e.g., build, test, and done. Both use an agile board to track progress as illustrated in Fig. 2. However, the methodologies differ in the events occurring between the product backlog and the customer (Linz, 2014; Saleh, Rahman & Asgor, 2017; Matharu et al., 2015).

Figure 1 The V-model software development process.

Figure 2 Illustration of an agile board.

Industry standards for functional safety

Among the standards for functional safety, the transportation domain is often considered important with respect to tools used during development (Asplund, 2014; Asplund, 2015; Asplund, El-khoury & Törngren, 2012; Conrad, Munier & Rauch, 2010; Ekman et al., 2014; Krauss, Rejzek & Hilbes, 2015; Notander, Höst & Runeson, 2013). IEC 61508:2010 (2010) is a generic industrial standard covering the lifecycle activities for systems in this domain. The standard also serves as a template for other standards. ISO 26262:2018 (2018) is the domain-specific adaption of IEC 61508 for the automotive domain (this, and many other standards, exist in several editions, and much literature instead investigate older version(s) such as ISO 26262:2011, 2011). EN 50128:2011 (2011) is the domain-specific adaption of IEC 61508 for railway control and protection applications. Derived from this standard is EN 50657:2017 (2017), which is an adaption of EN 50128 for application in the rolling stock domain. EN 50657 was partially created to ease work with non-safety related software after the changed definition of SIL 0 made in EN 50128:2011 compared to EN 50128:2001. The former definition of SIL 0 “no safety impact” was changed to “lowest level of safety impact,” rendering some confusion on how to handle products with no safety impact. EN 50657 therefore replaces SIL 0 with Basic Integrity (BI) for software that is not safety related (Nordström, 2017). Although more previous work has been done on EN 50128 when compared to EN 50657, we focus on EN 50657 due to its importance for the industry partner.

RTCA/DO-178C is a set of recommendations for compliance with regulations of civil aviation authorities, such as the Federal Aviation Administration (FAA) and the European Aviation Safety Agency (EASA). These guidelines are not derived from IEC 61508. The C-version was released in 2011 as the successor of DO-178B and simultaneously introduced DO-330 “Software Tool Qualification Considerations,” which provides guidance on tool qualification. DO-330 is very similar to DO-178C but adapted with objectives and requirements suitable to software tools (Rierson, 2017).

The safety standards use Safety Integrity Levelss (SILs) as a scale to classify risk and criticality, and the required assurance against failure. The position on the scale determines the integrity required to prevent failures and the mitigation measures required. The scales and contents of the levels are different between standards, but used in a similar way. IEC 61508 defines, from low to high level, SIL 1 to 4, and ISO 26262 similarly uses ASIL A to D (Ekman et al., 2014). EN 50128 defines SIL 0 to 4, a scale also used by EN 50657 but with SIL 0 replaced by BI (Nordström, 2017; EN 50657:2017, 2017). Finally, DO-178C uses Development Assurance Levels (DALs) E to A, a scale also used by DO-330 (Rierson, 2017).

According to Asplund (2014) and Notander, Höst & Runeson (2013), safety standards can be divided into two main groups based on their view on how trust in a tool shall be ensured. The first group focuses on means, where trust is established by generic measures such as thorough specifications and assessments during development of the tool, suggested or enforced by the standard. IEC 61508 and standards derived from it belong to this group. The second group focuses on objectives to be fulfilled, where trust in a tool is ensured by the applied constraints on its development process. DO-178 and DO-330 belong to this group but provides limited practical guidance on how that is to be achieved (Notander, Höst & Runeson, 2013).

Tool qualification

Tools may eliminate, reduce or automate processes in development of embedded systems. Malfunctions in the tool may lead to introduction of errors, or failure to detect errors, in the system. Therefore, tool qualifications or certifications are used to increase confidence in the tools. Qualification is sometimes required by standards. A tool certification can be defined as a complete set of activities to assert that an end product possesses a set of predefined characteristics, whereas tool qualification is a subset of these activities, ensuring that the confidence in the tool is at least equal to the confidence in the activities it eliminates, reduces or automates (Asplund, 2014). Tools are categorized according to the SIL of the tool or (sub-) system. The method of classification and different categories varies between the standards. IEC 61508, EN 50128, and EN 50657 all divide tools into either being on-line or off-line tools. On-line tools have a direct influence on the system during run-time and off-line tools do not. Tools categorised as off-line are then further divided into the three classes T1, T2, and T3, based on their potential impact on the system (e.g. a text editor is T1 because its output does not directly impact running code, but compilers are T3 because they do).

ISO 26262 instead classify according to Tool Confidence Level (TCL), based on determined Tool Impact (TI) and Tool error Detection (TD). TI is the possibility that a malfunction in the tool can introduce or fail to detect errors in the system and has two levels based on whether or not it can be argued that such a risk exists. TD measures the confidence in prevention from, or detection of, any shortcomings. If determination of TI or TD is not clear, estimation should be performed conservatively.

Software testing

In an embedded system, software is a major component, making software testing an important part of the development. The main purposes of software testing can be quality assessment and reduction of risk for software failures. Other typical objectives of testing are verification of specified requirements, validation of complete and correct functionality, enabling informed decisions with confidence in the quality level, verification of compliance with regulatory requirements or standards, or just feedback (ISTQB, 2011; Garousi et al., 2018, ISTQB, 2015; EN 50657:2017, 2017; Strandberg, 2018). To achieve efficient and correct testing many strategies, tools, and frameworks have been proposed over the years (Garousi et al., 2018).

Besides the actual execution of predefined test cases, the testing process includes activities such as planning, analysis, design and implementation of tests, reporting test results, and quality assessing the tested object. When execution of the component or system is part of the testing process it is referred to as dynamic testing, contrasted by static testing that only involve reviews of work products such as source code and requirements. The concept of quality assurance focuses on compliance with suitable processes to provide confidence in the achieved level of quality, and should not be confused with testing which is one of several inherent activities. Testing is a mean to achieve quality in different ways, while quality assurance deals with the entire process and is the enabler of correct testing (ISTQB, 2011).

By automating test execution with software, available resources can be utilized more efficiently, repeatability increases, costs decrease, and development efficiency improves. Test automation is, therefore, an important factor in agile development that enables fast feedback to developers and stakeholders, and it allows tests to be performed by a diverse pool of employees (Wiklund et al., 2017). Common concepts in agile development such as continuous integration (Stolberg, 2009) and automated acceptance testing (Haugset & Hanssen, 2008) heavily rely on test automation (Wiklund et al., 2017). For the implementation of test cases, monitoring and control of execution, and reporting and logging of results, it is necessary for test automation to involve the design of testware. This should include software, documentation, test cases, test environments and test data. The concept of test automation includes using purpose-built tools for control and setup, test execution, and evaluating differences between required and actual results (ISTQB, 2016).

Test automation of embedded systems may require a number of tools and a non-trivial flow of information (Strandberg et al., 2019). e.g., subtoolA may generate a test suite, subtoolB may initialize test cases one after the other, subtoolC may allocate the required subset of a test system, subtoolD communicates with each Device Under Test (DUT), test results are reported to a test results database (subtoolE) using subtoolF, and subtoolG is used to generate reports from the database (Strandberg, 2021). Based on the generic test automation architecture provided by ISTQB (2016) and the architecture at the industry partner (Strandberg, 2021), an example of a test automation architecture can be seen in Fig. 3. This illustrates a Test Automation Framework (TAF), which can be seen as a set of different tools with specific tasks that interact with each other.

Figure 3 Illustration of a test automation architecture.

Previous Work

This section presents previous work related to the problem of quality assuring software development tools, and position our work with respect to this. Also, approaches for increased tool confidence are identified and summarized.

Overview of related work

Several publications study approaches, challenges, and impediments related to combining plan-driven and agile methods. Notander, Höst & Runeson (2013) conclude that agile development can co-exist with plan-driven development provided that identified challenges are addressed. Heeager (2014) identifies nine practice areas of meshing methods from the different development processes. These areas are management strategy, customer relations, people-issues, documentation, requirements, development strategy, communication and knowledge sharing, testing, and culture. Documentation is determined to be the hardest, while requirements, testing and customer relations is considered difficult to combine. Development strategy, and communication and knowledge sharing were found to be combinable without impeding challenges. Heeager & Nielsen (2020) focuses on the four areas of documentation, requirements, life-cycle, and testing. Challenges and proposed approaches related to these areas are identified to enable understanding of possibilities and difficulties in performing safety-critical software development using agile methods. Hanssen, Wedzinga & Stuip (2017) outline an approach for extending agile methods, in particular Scrum, to achieve the objectives of the safety standard DO-178C (presented in ‘Industry Standards for Functional Safety’). The main idea is a distribution of the DO-178C process steps as sprints with the sequenced Scrum phases: preparation, development, and closure. Hanssen, Stålhane & Myklebust (2018a) present SafeScrum, a variant of Scrum which attempts to be a valid approach for development of safety-critical systems, based in compliance with IEC 61508. This is achieved by mapping Scrum activites to applicable steps in the V-model, while omitting system level risk and safety analysis, and validation, from the sprints. Ghanbari (2016) suggest that accumulated technical debt can be identified and managed, or even avoided, by utilizing agile practices in critical plan-driven software development. The author identifies that debt caused by e.g., requirement ambiguity, diversity of projects, inadequate knowledge management, and resource constraints may be mitigated by applying common agile practices such as small releases with continuous testing, iterative development, burndown charts and backlogs, and stand-up and review meetings.

Conrad, Munier & Rauch (2010) analyze differences in tool qualification or certification in transportation domain standards (we further evaluate this work in ‘Approaches for Increased Tool Confidence’). The authors conclude DO-178 to be the most stringent among the studied safety standards and emphasize the differentiation between development and verification tools, and that verification tools are less demanding to qualify. Regarding IEC 61508, they conclude that confidence in tool output should be achieved by certification when possible, but that the standard provides limited guidance on how to actually certify a tool in practice. When comparing DO-178 with ISO 26262, there are significant differences in how to conduct tool qualification. ISO 26262 has detailed guides on how to provide evidence that a tool is suitable for safety-related development.

Ekman et al. (2014) analyze qualification of existing tools as an alternative to the regular certification process provided by transportation domain standards (we further evaluate this work in ‘Approaches for Increased Tool Confidence’). According to the authors, tools used for development and test are commonly not developed according to the processes depicted in safety standards meant for certification.

Asplund, El-khoury & Törngren (2012) propose a method for qualifying software tools as part of tool-chains based on nine identified safety goals. The method is based on integration of tools in a tool-chain by using a hierarchy of organisation levels where lower levels are controlled by constraints from higher levels, thereby reducing complexity at lower levels. Using the reference workflow of Conrad, Munier & Rauch (2010) and the concept of Safety Element out of Context from ISO 26262, Asplund, El-khoury & Törngren suggest four steps for guiding and limiting the qualification effort. These include pre-qualification of both tools and tool-chain by representative use-cases and requirement deduction respectively. In a later publication, Asplund (2015) studies the relation of software faults to weaknesses in the support environments used, in relation to safety standards within the transportation domain. The author argues that standards often only concern tools in isolation which may lead to risks introduced by tool integration being ignored, a concern also raised by Conrad, Munier & Rauch (2010).

Positioning our work with respect to related work

Common for all identified publications on the subject of combining agile and plan-driven methods is the perspective of utilizing agile practices into an already existing plan-driven development process. Regarding studies of agile, traditional (plan-driven) or hybrid, there seems to be three common types of related studies (illustrated in Fig. 4). First (A), studying how to move development from a traditional approach towards a more agile approach. An example of this is presented in Hanssen, Stålhane & Myklebust (2018b), regarding development of safety critical software with the agile scrum approach. Another line of research (B) is to explore what agile software development is and how it is done; an example is a survey by Diegmann et al. (2018), where they identified that previous research on agile has focused on topics such as agile methods and practices; IT capability and agility; project, team and knowledge management; risk control and success factors; social interactions and behaviors; etc. A more recent line of research (C) is on hybrid methods, e.g., research by Kuhrmann et al. (2017); Kuhrmann et al. (2018) and Tell et al. (2021) in a large research project called Helena. They argue that most processes are hybrid, in the sense that they are traditional with some agility plugged in, e.g., they observed that a typical hybrid process is traditional in risk and configuration management, but agile in coding and testing. Furthermore, they identified that “these initiatives aim to bring more flexibility to processes…,” which implies that these methods somewhat overlap with research going from the traditional to the agile (A). Also, Tell et al. (2021) argue that “Traditional models are vanishing from researchers’ focus.” Related to these three strains of research, the article at hand (D) starts in an agile context and strives to go “backwards” in the sense that we try to explore what agilists could learn from traditionalists. We were unable to identify previous work incorporating plan-driven practices into an agile development process in order to increase confidence and quality in products and processes, which is the objective of this study.

Figure 4 Illustration of a three types of related work (A, B and C), and our study (D).

Approaches for increased tool confidence

Conrad, Munier & Rauch (2010) investigate standards to qualify two existing tools in accordance with ISO 26262. A directly extracted approach is to use a reference workflow from the existing tool (Ap.1). They identified the following steps for the qualification: requirements, specification, the model for code generation, generated code, and object code. Derived from this approach is the use of intermediate results in the chain of work steps to apply appropriate checks. The reference workflow is used to describe and limit tool use cases and lists available means for the detection of malfunctions and erroneous outputs. The reference workflow shall also describe verification and validation methods for each step in the workflow which may also identify means for error detection and prevention.

Wang et al. (2012) propose a semi-automated qualification method for verification tools that include hardware-in-the-loop test benches, for qualification of a new system or qualification after modifications. Their method is based on fault injection and monitoring (Ap.2), where faults are injected and the test system monitored for detection of the fault. According to the authors, applying the method on a new system requires the ability to run all test-cases both with and without fault injection. Failure of detection can be used to identify shortcomings in the testware of the verification tool. If no systematic faults are present in the testware, then one ought to analyze requirements conformance in order to identify design errors or insufficient requirements.

Ekman et al. (2014) propose approaches for tool qualification in the transportation domain. They target a tool for dynamic instrumentation based on binary modification. Several approaches were derived. First, to develop from scratch (Ap.3) by re-developing the entire tool or by constructing a complete safety case for the existing tool. Second, to qualify in accordance with a standard, e.g., by formally proving (Ap.4) that tool output conforms to specification, by automated correctness checks (Ap.5) of the tool output, by implementing a tool error detection system (Ap.6), or by applying design diagnostics (Ap.7) based on, e.g., Failure Mode and Effect Analysis to detect identifiable failures in the output. Their third approach is to design a protection harness (Ap.8) that detects and acts on errors in the tool, preventing them from propagating to failures. To implement a protection harness one has to consider the tool as a tool-chain of sub-tools (described in ‘The Test Tool as a Tool-Chain’ below). The protection harness is based on evaluating all intermediate results present in the tool-chain before letting the process proceed to the next step.

Hillebrand et al. (2011) propose a stepwise method tightly coupled to the V-model that we generalize to fit the scope of this article (Ap.9): (i) Describe all essential workflow steps with purpose and dependencies. (ii) Describe the used tool(s) and input/output for each step. (iii) Create and use requirement based checklists for each step to detect or prevent development errors. (iv) Break down the steps into use cases describing any user interaction, as well as different input/output or tool sequence scenarios. (v) Continue with identifying possible errors based on provided generic tool error types. (vi) Collect all previous steps in a checklist that includes detection/prevention/mitigation measures. Finally, the authors propose that the tool-chain structure (Ap.10) can be used to construct tests in a tool in order to detect errors by another preceding tool(again, see ‘The Test Tool as a Tool-Chain’).

Krauss, Rejzek & Hilbes (2015) evaluate requirements for qualification of software tools for hazard and risk analysis, that they compare with safety standards in the transportation domain. The authors provide three approaches: They suggest that development according to DO-330 life-cycle is a valid tool qualification method than can be used as guidance also in other domains (Ap.11). Secondly, validation by requirements-based testing (Ap.12). Finally, checks of completeness and correctness (Ap.13) of tool output should be achieved by a proper verification process.

Lloyd & Reeve (2009) report on their experience as assessors for certification according to IEC 61508. Their focus was on complete systems, but they provide lessons learned for both unsuccessful and successful cases that can be applied to tool development. Experiences from unsuccessful cases show that showing coverage at acceptance testing was not possible due to missing requirements specification or requirements that were not traceable through the lifecycle. The authors argue that structuring, tagging, and handling requirements can be made manageable by automating traceability (Ap.14) with a traceability matrix generated from a requirements database, or by using a requirement tracking tool. There was a lack of awareness and knowledge regarding static analysis techniques, with development teams not being aware of the benefits. The authors argue static analysis to be essential, with a need for several techniques such as i.a. control flow, data flow, range checking and unsafe code detection, and shared resource analysis. The authors argue that unit testing (Ap.15) should be preceded by static analysis and peer review, focusing on assumptions of pre- and postconditions. Difficulties in integration were found to often arise from defective or erroneously assumed module interfaces. They also emphasise the importance of configuration management and change control (Ap.16), and that reviews and issue tracking should be supported by workflow tools (Ap.17).

The main issue was legacy code, often developed over years without sufficient documentation. Bringing this code up to standard in retrospect would not be economically feasible. For small amounts of code, the authors recommend to re-develop from scratch in accordance with IEC 61508 (Ap.18). For large amounts of legacy code, they recommended to develop a monitoring and shut-down device for the main product (Ap.19), similar to the safety-shell mentioned by Ekman et al. (2014).

One approach for successful assessments mentioned by Lloyd & Reeve was to use a sequence of “mini-waterfalls” (Ap.20) for software releases with increasing capability, similar to combining plan-driven and agile development proposed by Hanssen, Wedzinga & Stuip (2017). Another successful approach is to invest effort into understanding the requirements (Ap.21) and knowledge-sharing by prototyping parts of the software. Other successful approaches mentioned are to use reviews in all stages of development, conduct research in tools and techniques and invest in training and development of good practices.

Asplund, El-khoury & Törngren (2012) and Asplund (2014) explore tool integration, i.e., automation supporting interaction between software tools or between tools and users in a tool-chain. They also survey four standards in the transportation domain. Asplund, El-khoury & Törngren and Asplund defines two models that are combined to identify risks and derive causal factors. First, the conceptual model, that consists of four levels focusing on risks related to tools and support environments which define how higher levels control lower levels. Second, the reference model (an extension of work by Wasserman (1990)), that describes aspects of tool integration by identifying relationships and borders for tool integration. The reference model covers five aspects of tool integration for supporting interactions—platform, control, data, process, and presentation. By combining the conceptual model and the reference model, and the risk analysis proposed by Asplund we identify ten safety-related characteristics of tool-chains that should be managed to mitigate risks (Ap.22). This approach includes: (i) Data integrity, to guard against internal data corruption and safeguard users from choosing bad artifacts. (ii) Data mining, to extract and present relevant information. (iii) Traceability, to know that the design supports the requirements and also how faults relate to each other if they combine to create a failure. (iv) Well defined data semantics, to allow users with different roles to understand each other. (v) Process notifications, for the tool-chain to notify users. (vi) Process control, the tool-chain shall provide automated process control, e.g., by checking for new versions and blocking or highlighting when something has been found to be erroneous. (vii) Customizable GUIs, to enable correct actions by users with different roles, knowledge, or expertise. (viii) Coherent time information, to enable correct comparison of artifacts from different systems, a global clock should be used. (ix) Automated tool usage, to avoid manual work when proceeding between tools. (x) Automated transformations of data, to avoid manual involvement in transforming data.

Notander, Höst & Runeson (2013) explore challenges in implementing agile methods in plan-driven development of safety-critical systems. The authors conclude that some of the main challenges are differences in documentation focus, tight collaboration with test-teams contrasted with requirements of independent testers, and that many small releases conflict with heavy certifications of each release. Complete requirements are central for the development of both safety-critical and non-critical systems and should be elicited by an iterative process. Traceability is mandated by safety standards and maintaining it may come with a high cost. However, maintained traceability can support agile and flexible development by identifying dependencies that need to be addressed during evolution. Having a clear and layered architecture with a generic bottom and building up with specific adaptions that cannot affect lower layers supports isolation of changes and minimizes re-certification needs. Derived from these insights were the following two approaches. (i) Construct requirements on tools used in the test framework that are elicited from, and traceable back to, the tested software and top-level functional requirements (Ap.23). (ii) Adopt a clear, dependency layered and continuously maintained architecture of the test framework where the potential impact of changes can be easily derived (Ap.24).

Wiklund et al. (2017) identify impediments related to automated software testing in general. They emphasize that development of a test tool is software development, and should be treated as any other software project and involve adequate treatment of standards, quality criteria, requirements, architecture, documentation, testability, and maintainability. Insufficient considerations of these factors may lead to poor test tool quality, and failure to detect defects. Low confidence in the test results may also lead to doubts whether failed tests are caused by the test environment or the tested software. The authors further identify the importance of ensuring that the environment is not difficult to use in a way that may lead to difficulties or confusion in performing or managing configurations. Tests executed on unknown or erroneous configurations can harm repeatability and impede detection of defects caused by unstable or misinterpreted results. We derived the approach to: Develop the test framework with at least the same rigour as the tested software (Ap.25), with special regards taken to address potential problems with performing or managing configurations (Ap.26).

In addition to approaches extracted directly, Hillebrand et al. (2011) also contained an approach that could be derived: the potential generic use of the proposed tool error types. They provide six basic error types for generic error classification (Ap.27) applicable to software tools: input errors, processing errors, process configuration errors, operating environment errors, misconceptions by user, and implementation errors by user. These generic errors do not provide mitigation strategies on their own, but may be suitable for use with the other proposed approaches to identify errors.

Common to the guidelines provided by IEC 61508:2010 (2010), EN 50128:2011 (2011), and EN 50657:2017 (2017) is that offline support tools shall be categorized into one of the three classes (discussed in ‘Tool Qualification’). For tools in the strictest class (T3), the standards list different types of evidence that can be used to show that a tool conforms to its specification or that failures in the output are detected. If a tool does not make direct or indirect contributions to the software under test, it will never be in the strictest class, but instead e.g., T2. According to these standards, “evidence listed for T3 may also be used for T2 tools in judging the correctness of their results.” Furthermore, tools shall be able to cooperate, such that output from one tool can be input for another.

The three main aspects of requirements for software support tools in IEC 61508-3 are degree of support for production of software according to requirements, clarity of operation and functionality, as well as repeatability and correctness of the output. Tools for stricter systems (T2 and T3) should have a specification or product documentation (Ap.28). Risks that these tools might affect executable software shall be determined by assessment, identifying failure mechanisms and applying mitigation measures (Ap.29). Other mitigation approaches are avoiding known bugs, restricted use of tool functionality, checking tool output, and using diverse tools. For the strictest applications (T3 tools), IEC 61508 suggests, as evidence for conformance: successful history of use (Ap.30) and validation (Ap.31). Or, if evidence is not available: effective measures to control failures (Ap.32).

EN 50657(c 6.7.4) addresses requirements on support tools in order to reduce the likelihood of introducing or not detecting faults during development. The standard mentions identification of potential failures (Ap.33) in tool output and measures to avoid or handle such failures. T2 and T3 tools shall have a manual or specification where tool behaviour, instructions, and constraints of use is defined (Ap.34). As evidence of conformance (for T3), EN 50657 provides more alternatives than IEC 61508, e.g.,: history of successful use (Ap.35), diverse redundant code for detection and control of failures (Ap.36), tool validation (Ap.37), compliance with SILs derived from risk analysis (Ap.38) of process and procedures, and other appropriate measures for avoiding or handling failures (Ap.39). If such evidence is not available, there shall be effective measures to control failures resulting from faults in the tool.

ISO 26262:2018 (2018) (part 8, ch.11) handles confidence in the use of software tools, with the objectives to determine the required level of confidence, and means for qualification when applicable. The main goals are to minimize the risk of systematic faults in the end product due to a tool introducing or failing to detect errors, and that usage of software tools does not affect compliance with the standard. The term “software tool” is deemed ambiguous, in the sense that it can vary from a single software package to an integrated suite of tools in a tool-chain, and also be applied to a variety of tools, such as commercial, open source, or in-house developed tools. As mentioned (in ‘Tool Qualification’ and ‘Overview of Related Work’), there are no distinctions made regarding how a tool is used or the possible effects on executable code as is the case for the previously mentioned standards. ISO 26262 states that requirements on the tool (Ap.40) shall depend on its role, related risks, and SIL. As internal prevention and detection measure (Ap.41) monitoring is suggested, and as external measures (Ap.42), guidelines, tests, and reviews. For verifying compliance to its evaluation, the standard suggests operating the tool with measures for error detection or prevention in combination with, e.g., fault injection (Ap.43) (similar to suggestions by Wang et al. (2012)). Also, verification of appropriate tool functionality in the user environment can be conducted by running a tool validation test suite (Ap.44). To ensure proper evaluation of usage, the standard suggests comparing outputs of redundant tools, performing tests, static analysis or reviews, log file analysis, and avoidance of problematic tool functionalities. The measures apply to both known and potential errors in the tool output. For evaluating the tool by analysis, prevention or detection can be achieved by redundant tasks or tools, or by rationality checks within a tool. Additionally, a tool can be used to verify the output of another precedent tool, implying a tool-chain structure.

If a tool is determined to have confidence level TCL 2 or 3, then qualification is necessary according to ISO 26262. For this procedure, the standard provides four different methods: (i) Validation, aimed at providing evidence for either absence of, or detection of assessed errors. From the method of validation, stand-alone strategies could be extracted as using a customized test-suite, and examination of reactions to anomalous operating conditions (Ap.45) such as foreseeable misuse, incomplete input data, incomplete update, and use of prohibited combinations of configuration settings. (ii) Increased confidence from use (Ap.46), requiring i.a. unchanged specification, sufficient data obtained from accumulated use, and malfunctions accumulated systematically. (iii) Evaluation of the tool development process (Ap.47), which should be based on an appropriate standard. (iv) Development in accordance with a safety standard (Ap.48), however “No standard is fully applicable to the development of software tools. Instead, a relevant subset of requirements of the safety standard can be selected.”

Summary of approaches from previous work

In this section, we have identified 48 approaches for increased tool confidence from previous work, enumerated in ‘Approaches for Increased Tool Confidence’. Through analysis, in order to identify similarities, we grouped them into 22 candidate solutions suitable for test framework quality assurance. Further, we identified four main groups for clustering: development, analysis, run-time measures, as well as validation and verification. These candidate solutions are presented with traces back to the original identified approaches in Table 1.

Research Methodology

This study was performed as an industrial case study (Runeson & Höst, 2009; Runeson et al., 2012), and the purpose of this section is to describe the essential elements of the case study design.

Rationale and purpose of the case study

The case study is being done to find and validate the candidate solutions, both from the literature and an industrial viewpoint. The secondary rationale is to assess the applicability of the candidate solutions with regards to quality assurance of frameworks for automated software testing. The expected outcome of this case study is the confirmation that strategies for increased confidence and quality regarding tools used for automated software testing in non-safety development can be found or created from concepts and strategies related to safety-critical development, while maintaining agile and efficient processes.

Case and units of analysis

The industrial partner in the case study, Westermo Network Technologies AB (https://www.westermo.com), specializes in industrial communication equipment for domains with high demands on robustness and availability, such as train, oil and gas, maritime, and water treatment. Thus, many customers have to comply with a functional-safety standard, which imposes demands of high quality on products acquired. Different devices for robust data communication are developed, e.g., robust Ethernet switches. Each device is an embedded system, running the Westermo Operating System (WeOS), developed at Westermo. While based on GNU (https://www.gnu.org/) and Linux (https://www.kernel.org), WeOS also includes other open-source software libraries and proprietary code. This accumulates to a source code base of millions of lines of code.

To ensure the quality of the products, Westermo applies automated testing, conducted on several test systems each night. Further, there is risk-based testing, where identified risks are used to conduct manual testing or to construct new test cases, as well as release testing using third-party robustness and performance tools in combination with reviews.

A test framework has been developed, implemented and maintained over several years. The framework consists of testware and different setups of devices into several physical test systems with varying layouts, each containing four to 25 devices with hardware, firmware and software. The in-house developed testware is used to configure and control the devices, which are running some version of WeOS. Further, the testware contains all test scripts, configurations and procedures, and is also used for activities surrounding the tests such as test case selection, setup, tear-down, and logging. The framework allows for both manual and automated testing, simulating installation scenarios and hardware/software combinations to test e.g., a software feature, a physical device, or a customer-specific case (Strandberg, 2021).

The studied case in the research is defined as the industrial partner and the products developed. The unit of analysis is defined as the development and maintenance of the test framework, utilized at the industrial partner for the execution of manual and automated tests of produced products.

Case selection strategy

We use the rationale for single-case designs from Yin (2009) to motivate the case selection strategy. According to Yin, a single-case design is justifiable “when the case represents (a) a critical test of existing theory, (b) a rare or unique circumstance, or (c) a representative or typical case or when the case serves a (d) revelatory or (e) longitudinal purpose”.

One rationale for the use of single case is that it is suitable for confirming, challenging and extending the propositions derived from theory. The circumstances in which the propositions are applicable are met by our selected case, i.e., software development of an operating system that is a key part of the embedded system. A second rationale for a single case is when the case is an extreme or a unique case, however, this does not apply in our context, rather our case is representative or typical (third rationale). The industrial partner and the products developed are typical of many embedded system manufacturers, both in the same domain of industrial communication equipment and in other similar domains. Thus, the results taken from this case are assumed to be typical of the experiences of an average embedded system manufacturer. The fourth rationale concerns if the case is revelatory, meaning that it has previously been inaccessible to scientific investigation. While our case is not completely revelatory, the context of the investigation nevertheless makes the results interesting since we were not able to find research investigating lessons to be learned from plan-driven safety critical development to agile software development. The fifth rationale for a single-case study concerns the use of a longitudinal case, but this is not applicable in our context.

Theory

This case study is based on the foundations of a rather extensive literature study, where 48 approaches for increased tool confidence from previous work have been identified. We have grouped them into 22 candidate solutions suitable for test framework quality assurance. This literature study forms the basis of answering the first research question, while the second research question addresses the applicability of identified approaches for test framework quality assurance.

Research questions

Together with the industrial partner and for the case-specific test framework, we formulated two research questions.

RQ1: Based on the approaches proposed in relation to relevant safety standards, what strategies for increased confidence in software tools can be found or constructed?

RQ2: Which of the above strategies are applicable regarding quality assurance of frameworks for automated software testing?

Propositions

There are two propositions that underpin our case study. First, that certain strategies for quality assurance of software tools could possibly be constructed from approaches gathered through theory. Second, the practicality of such strategies could be established via quality assurance of frameworks for automated software testing. These two propositions are directly linked with the two research questions, RQ1 and RQ2.

Concepts and measures

The main concepts and measures were described as following.

Approaches: These are methods of quality assurance of software development tools in functional safety development and agile development.

Candidate Solutions: These are refinements to the approaches found from theory to establish test framework quality assurance.

Applicability of approaches: The applicability of approaches was measured both qualitatively and quantitatively. Qualitative indicators included: good idea ‘:-)’, bad idea ‘:-(’ or indifferent opinion ‘:- |’, whereas quantitative indicators showed percentage of effort the focus group would like to invest in the approaches.

Methods of data collection, data analysis and data selection strategy

Two methods of data collection (literature study and focus group) were used in this study. The data collected from the literature study was analyzed qualitatively, while the data from the focus group was analyzed both qualitatively and quantitatively. Further details on the two data collection methods are presented in their dedicated sections below.

Literature study method

The literature study was based on guidelines on literature studies and snowballing (Kitchenham & Charters, 2007; Wohlin, 2014). The process started using an initial set of articles identified using Google Scholar or authors’ prior knowledge (Shahin, Babar & Zhu, 2017; Asplund, 2014; Notander, Höst & Runeson, 2013; Ghanbari, 2016; Conrad, Munier & Rauch, 2010; Garousi et al., 2018; Mårtensson, Ståhl & Bosch, 2016; Strandberg et al., 2019; Wiklund et al., 2013; Zhi et al., 2015). Articles were included if they (i) discussed tool qualification in relation to a safety standard, (ii) covers challenges related to test automation, tools or frameworks, or (iii) covers challenges in combining safety-critical plan-driven development with agile processes. For backward snowballing, the reference list of an already included publication was studied to identify additional publications to be included. For forward snowballing, citations were identified in the later publications back to an already included publication. Citing and cited publications were then evaluated for inclusion or exclusion. Further, to find missed clusters of publications, additional searches were performed in parallel to the snowballing process. From the articles, we identified approaches based on two criteria: (C1) approaches that could be extracted directly from the article, or (C2) approaches that could be derived from the article. Based on the initial set of articles, a total of 32 articles were processed, and nine included based on the inclusion criteria—three of which were in the initial pool of articles.

The literature study further included the review of three standards used in the transportation domain, specifically the sections/clauses addressing software development tools, which gave a set of approaches additional to C1 and C2. The relevance of these standards is motivated in ‘Industry Standards for Functional Safety’, and therefore no further inclusion criteria were applied. The approaches were further analyzed to identify similarities in concepts and to merge duplicates into candidate solutions. A candidate solution is a principle or practice derived for increasing quality of and confidence in an automated software test framework. Four main groups were identified: development, analysis, validation and verification, as well as run-time measures ‘Summary of Approaches from Previous Work’.

Focus group method

The method used for conducting the focus group was based on guidelines presented by Morgan (1996) and a literature study on focus group methodologies by Hylander (1998). The focus group was a self-contained activity containing both a qualitative and a quantitative part. In order to prepare the participants, we first introduced the purpose and structure of the focus group, the concept of a candidate solution, and the tool-chain concept (see ‘The Test Tool as a Tool-Chain’). Qualitative data was collected through moderated group discussions structured according to the four identified main aspects of candidates; development, analysis, run-time measures, as well as validation and verification. Each aspect was initiated with a short free discussion based on an open question on the subject of the current aspect. The objective of this activity was to have the participants introduced to the main subject and warmed up to get the correct mindset before presenting the candidates for further discussions. A discussion guide to stimulate discussions, if needed, had been prepared. The candidates in each group were then presented individually and accompanied by more detailed examples and/or considerations regarding the specific candidate. As base for discussions, the same three open questions were used for all candidates regardless of group:

– What would this concept mean in the context of this company?

– Is it a good idea?

– Why is it, or is it not, a good idea?

This process was iterated four times, once for each aspect, thereby covering all main aspects and the candidates.

The objective of the quantitative part was to obtain an indication of the perceived value of the candidates and create a perception of prioritization for further and future work. The method used was inspired by Planning Poker (Grenning, 2002) and the Delphi-method (Dalkey & Helmer, 1963). The participants were asked to imagine having 200 man-hours to invest in candidates of their choice. They could choose to invest all in only one candidate or to distribute their investment over several candidates. All candidates were then collectively presented and the participants were given time to reflect individually and write down their answer. Finally, the participants presented their choices and motivations one by one, and the results were simultaneously summarized and presented in a spreadsheet for all to see. The idea with this procedure was to stimulate discussion over choices and motivations. The focus group ended with a summarizing event, asking the participants if there were any candidates they had expected to be presented but that were missing, and if they could share any other thoughts or ideas regarding the material that had been presented to them. Due to the Coronavirus outbreak, the focus group was executed as partly remote with some participants on link. The presentation was simultaneously displayed physically and shared over the tool used to host the remote meeting.

Participants: Aiming at diversity in terms of experience and specialization, we recruited a stratified convenience sample of six individuals for the focus group: one manager for the software test team who is responsible for the framework, one manager for the WeOS team, three developers from the test team, as well as one developer from the WeOS team.

Execution Roles: The focus group was driven by three execution roles: (i) The first author was responsible for preparing and running the presentation, and further to introduce and explain presented activities, concepts, and candidates. (ii) The second author acted as moderator during discussions, i.e., by keeping track of coverage regarding both topics and speakers, and asking for further elaborations when necessary while trying to maintain fluent and self-driven conversations. (iii) The third author acted as support of the execution by clarifying the purpose, assisting in understanding the purpose, and step in if needed to keep the activities in line with the goals.

Data Collection: All qualitative data collection was conducted by taking notes during the discussions, instead of making recordings for later analysis. As suggested by Krueger & Casey (2014), recordings are not mandatory for data collection, since analysis can be performed on the basis of memory and/or notes alone. To mitigate risks of missed or misunderstood discussions as well to ensure capture of relevant notes, three authors took individual notes simultaneously. The notes were later shared to consolidate the findings by comparing them against each other.

Data Analysis: The first step of analysis was to merge the handwritten notes taken by all three execution roles. The merged notes were then further processed and summarized, removing duplicates, clarifying expressed opinions by the collective notes on the same subject, and identifying which participant had made what statements where such coding was missing in some of the notes. The resulting merged and processed notes where then analysed in the context of the group being the unit under analysis instead of the individuals of which the group consisted. Group opinions were differentiated from individual opinions by attempting to identify consensus reached within the group. Further analysis attempted to identify and understand which comments were reactions to direct questions, and which were spontaneous reactions to the ongoing discussion between participants. The results of the analysis is presented in ‘Qualitative Results of Focus Group’. Notes regarding the quantitative part where analyzed using the same process as described above to allow comments and motivations during these activities provide for a deeper understanding of the results, which are presented in ‘Focus Group Quantitative Appraisal’.

Quality assurance, validity and reliability

While there was no pilot study conducted to evaluate the case study design, it was reviewed by three experienced researchers with several years of experience in conducting industrial case studies, especially one of them having a split role in the case company and academia. The design activities in the case study were performed together with these experienced researchers, thus the activities were planned and discussed beforehand. In order to ensure quality, validity and reliability of the focus group feedback, a discussion guide was prepared and each aspect was explained with detailed examples. The same three questions were used to get feedback for all the candidates and the process was iterated four times, once for each aspect. Furthermore, the quantitative results of the focus group were summarized and presented in a spreadsheet for all to see. Lastly, the focus group ended with a summarizing event, with an opportunity given to the participants to confirm their feedback and to fill any missing information. This yielded the identification of three additional candidates, as presented in ‘Additional Candidates Identified by Focus Group’.

Validation of Candidate Solutions with Focus Group

In this section we describe the validation of the candidate solutions, i.e., preparing and conducting a focus group. The results are presented as both qualitative and quantitative outcomes, complemented by additional suggestions of candidates.

Focus group preparation

From previous work and standards, 48 approaches for increased tool confidence were identified in ‘Approaches for Increased Tool Confidence’. These were groups into 22 candidate solutions suitable for test framework quality assurance, clustered into four main groups: development, analysis, run-time measures, as well as validation and verification. Table 1 lists these candidate solutions as presented during the focus group, with traces back to the original identified approaches.

The test tool as a tool-chain

A meta-approach common in the literature is to see the test tool (e.g., a test framework), not as one entity, but as a tool-chain built up of the tools of the framework, (see Fig. 5). The reference workflow from Conrad, Munier & Rauch (2010) and workflow steps from Hillebrand et al. (2011) are based on a flow through a chain of tools, where use cases with possible errors, validation and verification means, as well as failure mitigation measures are applied to each step through the chain. This approach is also supported by, e.g., using the tool-chain to detect errors (Hillebrand et al., 2011), the safety shell approach (Ekman et al., 2014), the importance of tool integration emphasised by Asplund (2014), that tools shall be able to cooperate (IEC 61508:2010, 2010; EN 50657:2017, 2017), and that a tool can be “a suite of software tools integrated into a tool-chain” (ISO 26262:2018, 2018). In practice this could be understood as an automated test tool-chain consisting of several different tools, performing different tasks, that as a whole result in a complete test framework. For each individual tool in the tool-chain, different approaches are suitable depending on the nature of the tool and the task it performs and should be applied accordingly. Therefore, a basis for interpreting, understanding and applying the proposed candidate solutions, presented in Table 1, is to view them through a “tool-chain lens.” In particular, what is an individual tool in a tool-chain at one level, can be seen as separate tool-chain when evaluated closer. Different levels of tool-chains may exist depending on the complexity of the system. The idea that a tool can be a tool-chain when looking at the inherent parts is supported by the definition of a tool in DO-330, as quoted by Rierson (2017): “A software tool can be a complete program, or a functional part of a program.” The tool-chain model also implies that if classification based on the possibility to introduce errors or fail to detect them, is to be performed in accordance with an applicable standard, the classification should be applied to each individual tool in the chain based on analysis of the specific tool where possible errors are identified. Analogously, determination of Tool Confidence Level could be performed on each individual tool. Assessment of the complete framework could then be derived from motivations applied for the classification of each individual tool, aspects considered for integrating the tools, and the results on framework level of failure mitigation measures applied for each individual tool or interactions between tools. Thus, confidence in the complete framework should be argued as the sum of measures applied to sub-tools, and the confidence in their results and interactions.

Figure 5 Conceptual visualization of a framework tool-chain model.

Qualitative results of focus group

The qualitative results of the focus group are presented based on the main aspect of the candidates (development, analysis, run-time measures as well as validation & verification). Interpretation of qualitative results, based on discussion analysis, was performed by applying a three-step scale: good idea with high value, indifferent or ambiguous opinion, or unappreciated idea with little or no value, as presented in the Qual.-column in Table 1.

Development

The introductory discussion was based on the question “thinking back, do you know of any events, positive or negative, that could be linked to the development process?” The answers tended to focus more on negative aspects, with mentions of a rapid development pace leading to missed test results, or even no results at all, after updates or to the testware. Adding tests in simulated environments and more extensive reviews was argued to be potentially beneficial in this aspect. Extending the development process with added phases was also mentioned with considerations of cost and productivity, and how to gain the best effect. Developers experienced that sometimes tests were missing, requesting testing of the tests. The focus group also emphasised differentiating between development and production environments.

D.1 Apply measures to avoid development faults introduced by misconceptions. The focus group found this to be a good idea, they suggested to clearly define what a review is, and what is expected during the review. To emphasize the importance of documentation to be understood by different people and after long periods of time. To use checklists as a mean to achieve clarity. To have a clear architecture in order to easily see dependencies and the effect of changes. To conduct analyses of errors to gain statistical data and derive the root cause to avoid similar issues in the future. They also suggested that making complete predictions on potential faults is difficult, and wondered whether FMEAs would be applicable to mitigate this.

D.2 Apply restrictions on tool usage. The focus group had mixed opinions, and felt that applying these types of measures initially had a low priority. However, benefits could be seen regarding third-party software with known issues, and that a policy on what parts of a tools to use for a specific purposes, and what functionalities to avoid, could be beneficial.

D.3 Apply measures to avoid potential errors introduced by users. The focus group found this to be a good idea. It could be beneficial in the aspects of masking complexity for the users, and also minimizing manual configurations to the greatest extent possible. Complexity that grows over time can result in mistakes which could lead to lost test results.

D.4 Develop the test framework based on requirements. Here the focus group had a mixed but mostly negative view. A higher focus on requirements is a reasonable approach from a long-term perspective, since it could yield a more testable and correct product. However, too high focus on requirements could have a negative effect if the requirements are not complete, thus giving rise to missed aspects. Clarity in requirement elicitation and ownership is important.

D.5 Apply measures of rigour to the development process. The focus group found it reasonable and beneficial to apply the same test strategy on the test framework as what is conducted regarding the software to be tested. They consider extending the framework development process to include more unit tests. Also, they argued that one needs to determine a reasonable level of quality assurance and rigour in the context of the test framework and integrity of produced test results.

D.6 Re-develop the entire test framework in accordance with a suitable safety standard. The focus group argued that this was not applicable due to e.g. the high amounts of waste and significantly increased costs, and that this would not necessarily yield any increased quality.

Analysis

The introductory activity based on the open question “what are your thoughts on analysis to identify potential problems in advance?” was mainly positive. The focus group saw benefits in focusing efforts in advance. They saw value in being able to determine effects of changes in advance, and gave examples of difficulties with current tools that could benefit from more analysis before deployment. They also mentioned difficulties in capturing all possible events, in identifying events that may never actually occur, and the importance of keeping analysis at a reasonable level. Further, they discussed the importance of performing root-cause analysis when errors occur, in order to identify proper measures for avoiding similar errors in the future.

A.1 Perform formal risk and impact analysis. The participants were positive to this approach, in particular to risk-based testing. They also argued that by using the same approach for test framework work as with any other development, this could also yield enhanced cooperation, communication and understanding.

A.2 Analyze the tools using a tool error checklist. The focus group interpreted this as Definition of Done (DoD),3 with general aspects and measures to be assured. A benefit could be to not miss relevant activities, but the focus group has a hard time imagining how to create generic checklists from a risk analysis.

A.3 Perform analysis with regards to abnormal operating conditions. The participants saw this approach as having great value, and made references to historical events where this could have been useful. Errors of this kind should be analysed for similar potential events to determine counteracting measures.

A.4 Analyze using detailed peer-reviews during development. The focus group were very positive to this approach, and in the quantitative appraisal this was one of their favourites. They saw it as highly important with potential to create a basis for many developing benefits, such as, definitions of what to look for, knowledge sharing, the low cost compared to the introduction of faults in the product, and decreased risk of potential errors in the product. They saw great value in pair-design and pair-programming as review a method, as well as presenting your solution to someone else. However, the phrase “detailed” should be clarified, interpreted as the specification of review execution, included activities, and expected outcomes. Also, this approach could potentially block development progression if reviews are not prioritized, and there is a risk that reviews become just a “tick-in-the-box.” A.5 Analyze the tools with static analysis. In general, the focus group saw static code analysis as a good idea, but the value of analysis tools should be evaluated for each specific case. They had positive experiences of linting4 tools as a method of performing static code analysis. They mentioned that false positives created by a tool could render a lack of trust in produced results over time.

Run-time measures

The introducing discussion was based on the question “what is your spontaneous interpretation of a run-time measure in the context of the test automation framework?” The conversations mainly revolved around measures to avoid potential problems, such as overloading a server, full disks, and no access to databases. Current implementations, such as redundancy in writing to a database, were also mentioned.

R.1 Develop automated sanity checks of important tool actions. The focus group felt that this was mostly redundant if risk-based testing is correctly introduced, except for dynamic aspects of the framework. The group mentioned that it is important to verify the correctness of the environment preconditions before testing. Historically, an error in one test-suite has sometimes led to the failure of several sequential suites, which potentially could be mitigated by ability to reset the system upon failures and then start at the next step.

R.2 Implement checks of output from a preceding tool conducted in a subsequent tool in the tool-chain. The participants mentioned that this could be hard to implement, since many things could potentially go wrong, but it should be possible to determine and exceed a minimum level of appropriate checks. The discussions focused the value of assuring that the correct conditions exist from the previous step in the current context. By having this in place, a benefit could be to more easily distinguishing between errors in the software under test and the testware, since incorrect conditions for a test could be misinterpreted as an error in the tested software.

R.3 Develop a monitoring system for error detection and prevention. The focus group saw this as a good approach, and discussed work on historic issues. If there is a lack of history of the data-flow chain then this could impede troubleshooting of errors. The focus group speculated about the benefits of visualisations in a global log management system to which all tools/subsystems could report their status and problems, and compared this to Lauterbach (https://www.lauterbach.com) and Jeager (https://www.jaegertracing.io). They saw clear benefits to monitoring and notifications of test progression, especially during the final testing at release-time.

R.4 Develop protection against identified abnormal operating conditions. The focus group requested that test execution could be halted if errors were detected. Such that these could be resolved before continuing, and that a failure in a test should not affect subsequent testing. The group desired the ability to reconfigure a physical test system and the included tests in the event of a lost part of a test system, and that the testware should automatically restart certain services. The group also mentioned the potential use of AI to analyze sequences and find problematic patterns and then trigger a reset, thereby allowing the suite to continue without errors. The focus group were of the opinion that detected errors should be cherished as a potential source of information and that it could lead to improvements.

R.5 Implement redundancy in tools and tool-chain. For the focus group it was unclear how to interpret redundancy in the context of the test framework, e.g. does unequal multiple test systems constitute redundancy, and is the purpose to have availability or correctness? Also, work on redundancy was ongoing, e.g., implementations with Kubernetes (https://kubernetes.io) and Docker (https://www.docker.com) with supervised and distributed test resources that implies redundancy.

Verification & validation

The opening discussion on the question “what comes to mind when thinking about achieving confidence in intended behaviour?” brought up that confidence is the outside experience of the framework. Responsibilities to write sufficient tests lie on the software developers, and to facilitate the tests lie on the test team developing and maintaining the test framework. Trust in the produced test results is essential to avoid e.g., developers being reluctant to question their implementation and instead argue for errors in the framework when a test fails.

V.1 Utilize a suitable safety standard to validate the tool and related processes. The group felt that being influenced by a safety standard may be good for some specific problems, but utilizing a complete standard for the test framework is not relevant as long as the tested software is not considered safety-critical.

V.2 Formally prove that tool outputs conforms to specification. The focus group argued that it is crucial to provide evidence of correct functionality for company-specific tools. e.g., the performance of the case company’s regression test selection tool (Strandberg et al., 2016), an in-house solution anchored in years of research and crucial to the applied test strategy.

V.3 Base tool confidence on history of successful use. At first, the focus group argued that this was not applicable given the frequent code changes of their internal testware. However, this approach was seen as applicable to third-party solutions as a mean of resource management, spending less time on tools where confidence already exist. The group emphasised that this is a valuable approach when selecting new third-party solutions to build into the testware.

V.4 Create a customized tool validation test suite for all use cases. The focus group saw it as valuable to identify a subset of critical use cases to validate intended behaviour, but objected to the phrase “all”, since they saw it as unreasonable to identify and test all possible use cases.

V.5 Perform tests based on fault injection. The participants saw this as a small, and relatively easy approach to implement, with potential to generate significant value—and that this could help developers in understanding how robust the system actually is.

V.6 Perform unit tests on all modules and tools in tool-chain(s). The focus group saw this as a very valuable approach, and this was also one of the most liked approaches in the quantitative appraisal. However, the focus group also objected to the phrasing “all”, as it is not reasonable and also potentially costly to perform.

Focus group quantitative appraisal

During the focus group, the members could vote for the candidates they preferred (as described in ‘Focus Group Method’). The development candidates received the least interest with 13% of votes, whereas the other three groups were about as popular with between 25 and 33%.

Derived from comments and motivations during the quantitative activity were the following primary insights. Establish a baseline to define a lowest bar of acceptance where guidelines and checklists for reviews are important means to achieve a unified view of how reviews are conducted; what is included in a review, and what development artifacts should be reviewed. One suggestion concerning checklists was to create a proposal for a DoD. Unit tests are important, especially combining unit tests with Continuous Integration and possible implementations in staging environments. Monitoring is important to help derive where an error has occurred and enable alerts of errors to provide awareness. Root cause investigations were emphasised with proposals for error investigation commissions, extended root-cause analysis and issue tracking. Further, the group expressed an expectation for requirement-based testing to be explicitly stated as a candidate. The importance of durability over time and scalability were also emphasised.

Additional candidates identified by focus group

From the final summarizing event, it could be determined that overall, the candidates were perceived as valuable and a suitable base for further discussions. During these discussions three additional candidates were identified: First, to implement requirement-based testing. This candidate was found in the literature study, but had, by mistake, been overlooked. Without the focus group, the mistake would probably not have been discovered (V.7 in Table 1). Second, to perform sufficient root-cause analysis on detected errors. The focus group suggested to, e.g., initiate error investigation commission, perform post-hoc analysis on occurred failures, or to utilize a tool for issue tracking (A.6 in Table 1). Third, to halt execution on detection of errors or erroneous conditions. Sometimes it was deemed relevant to pause the test execution instead of continuing with the next test-case in current suite (R.6 in Table 1).

Summary of results and final candidates

Table 1 contains the final refined candidates and summarizes the associated results from the activities of the focus group in the right columns. Due to its design, the quantitative part did not further address any of the unappreciated candidates or other negative aspects. Therefore, all candidates included in the result of the quantitative part can be considered perceived as good with value bringing aspects. Additional candidates derived from the summarizing discussion were added and treated equally to the qualitative results. The candidates have been rephrased in accordance with the focus group results.

From analysis of the collected data, aspects and concepts that were repeatedly mentioned in different contexts during discussions were identified. These can be summarized as follows: (i) Measures intended for increased safety does not necessarily entail increased quality. (ii) Quality assurance and rigour applied regarding the test framework has to be reasonable in relation to the tested software. (iii) Confidence in results created by the framework from all stakeholders is very important from several perspectives. (iv) The required cost and effort have to be in balance with the expected gained effect. (v) A baseline should be established by setting a lowest bar of acceptance. On the more practical side, there were also reoccurring discussions, summarized as: (i) The expected content and execution of reviews, documentation, and similar activities has to be clearly defined. (ii) The environmental and other conditions regarding the execution of the test framework must be sufficiently ensured. (iii) Errors related to execution of a test-case cannot be allowed to have any effect on subsequent test-cases or test suites. (iv) It is important to be able to distinguish errors in testware from errors in the tested software.

Also, comprehensive root-cause analysis upon detection of occurred errors were repeatedly discussed as important to identify other similar possible errors. In ‘Approaches for Increased Tool Confidence’, only chapters related to tool qualification/certification were included. These chapters did not reveal any similar concepts.

Discussion, Threats and Future Work

In this section, we summarize and discuss the results of the research questions. Later, we also discuss the threats to the validity of the study as well as the future work.

Strategies for increased confidence in software development tools (RQ1)

Through a literature study targeting both safety standards and related work, we identified 48 approaches for test framework quality assurance ‘Approaches for Increased Tool Confidence’, which after refinement and validation resulted in 25 candidate solutions (three of which were added as a result of the focus group) (Table 1). The analysis of the literature identified that, as a basis for interpreting the candidates, the tool or the framework should be seen as a tool-chain build up of sub-tools and tasks—a point of view highlighted by e.g., Asplund, El-khoury & Törngren (2012); Asplund (2015) and Ekman et al. (2014). Depending on the nature of the sub-tool/tool-chain and the task it performs, different approaches may be suitable. Identifying interaction sequences enables for tests to be written at an early stage, as soon as there is access to intermediate results, instead of later testing the entire framework from a black-box perspective. When applying a standard, the inherent sub-tools and tool-chains can be classified on an individual basis and confidence argued as the sum of applied measures to individual parts and the integration between them. By proposing to do separate classifications of sub-tools, we extend the findings of Ekman et al. (2014) and Conrad, Munier & Rauch (2010). However, this is not aimed at dressed up classifications, but rather to enable a more efficient resource management and focus of efforts.

Combined with this insight, the candidates constitute a list of general measures, in four aspects: development, analysis, validation and verification, and run-time measures. For industrial practitioners, the candidates may provide guidance by proposing activities for quality assurance of in-house tools. It is also possible that subcontractors to companies in the safety-critical domain may find the results valuable, e.g. through facilitated communication and understanding concerning audits, etc.

Applicability and practicality of identified strategies (RQ2)

The implications and perceived industrial value of the refined candidates were evaluated in a focus group, conducted in collaboration with the industry partner. The focus group perceived that measures applied for increased safety do not necessarily lead to higher quality, and that the level of rigour applied on a development tool has to be reasonable in relation to that of the tested product—there has to be a balance between cost, effort and gained effect. The focus group highlighted that it is important to set a lowest bar of acceptance, and that the expected content in reviews and documentation has to be clearly defined. Also, it is important to ensure correct conditions in the tool environment, and to have the ability to differentiate between errors in testware and tested software. Finally, errors in one test case cannot be allowed to affect subsequent tests or suites. These insights can be considered to complement research on shifting plan-driven development towards agile processes, e.g. previous work performed by Notander, Höst & Runeson (2013); Heeager (2014) and Heeager & Nielsen (2020), by providing aspects from the opposite perspective.

The candidates were evaluated qualitatively and quantitatively (Table 1). The unappreciated candidates were those entailing the most effort where little or no gain could be seen. For several of the candidates considered as high value the discussion involved historical or current events. This result also provides information that indicate where initial efforts should be placed, which could be potentially be utilized in other industrial contexts than the case-specific. In addition to validation of identified candidates, the focus group also proposed additional candidates perceived as missing(which led to the rediscovery of a candidate lost in the process).

In the first appendix of the supplementary material that we attach with this article, we suggest a possible application of the results as an augmented agile process inspired by mini-waterfalls: development in isolated entities with added rigour through mini V-models controlled by DoDs. This is intended to be applicable, not only for development of software tools in particular, but to any software development in general. This process was made case-specific by defining the content of the DoDs which control the transition between phases.

Threats and limitations

The process of extracting data in the literature study was performed in a subjective way and may have been biased by prior education and existing knowledge. The size of the initial set of included publications could be perceived as inadequate. This was partially addressed during the study by performing searches for new publications in parallel with the snowballing process. It can also be argued that the extraction of data led to concepts being taken out of their contexts and presented in a subjective way. First, in the process of merging concepts during the analysis of the literature, we increased the level of abstraction of the candidates and applied a context specific for the industry partner. Also, candidates presented in different main aspects often have a sequential dependency where they build on each other, making it unfeasible to cherry-pick candidates perceived as adding the most value. Finally, the identified candidates depend on the relation to the presented perspective of tool-chains, meaning that existing and future tool-chains in the framework has to be identified to derive practical implications.

One threat related to the focus group is that we only used one group, and only performed one session. Having only one group eliminates the possibility to compare results and detect anomalies or misconceptions. However, it could be argued that the participants’ perception of the candidates was to some extent validated by the quantitative part at the end of the focus group session, where any major misconceptions would have been picked up and rectified. Involving a larger number of participants would most likely have given a greater sample size and more diverse feedback. But in terms of diversity of roles participating in the focus group, only a few relevant ones were omitted, e.g., no software architects were present but they typically have similar competences as senior developers. Performing only one session also eliminated the possibility to alter the questions and the structure of the focus group if shortcomings had been discovered. Having an on-line session may have affected the discussions since most non-verbal communication is presumed to have been lost. Furthermore, one participant had to leave before the session was completed. Overall, it was sometimes hard for the participants to stay on the specific subject of a presented candidate during the discussions.

Case studies do not often claim strong generalizability. This study is no exception as it is possible that our findings may not be completely applicable in different contexts. e.g., the candidate A.4 Analyze using well defined peer-reviews during development, was one of the most favored among our participants. However, in another context where there could be very few developers, or developers who are inexperienced, it might be difficult or impossible to conduct reviews due to lack of system understanding. However, despite this limitation, other researchers like (Briand et al., 2017; Hevner et al., 2008), have found industrial case studies as being valuable. A concern regarding the suggested DoDs (see first appendix), is the influence from already existing DoDs related to WeOS development on e.g., the identified phases. Also, the validity of the DoD activities is dependent on the quality of support documentation that may not yet exist.

Revisiting related work from the perspective of the focus group

Most hybrid development models are either combinations of different agile practices, or start as traditional models with agility plugged in. The models are based on experience collected over time and changes are typically not, as one could perhaps expect, driven by company size, domain or external standards (Kuhrmann et al., 2017; Kuhrmann et al., 2018). In their 2018 book, Hanssen, Stålhane & Myklebust (2018b) propose an incremental safety critical software development process. At the core are two parallel backlogs, one functional product backlog and another one for safety; as well as rigorous traceability between artifacts, and separation of roles into teams and a dedicated team for safety. One obvious advantage is, of course, that feature growth can be incremental (instead of specifying all of the system before implementing the first line of code). An important difference is that their model is used for developing safety-critical systems, and moves from the traditional towards the agile, whereas there are no strict requirements on safety for our model, and we move “backwards” from agile to traditional.

The most important candidate for the development aspect with 13% of the quantitative appraisal was D.1: to apply measures to avoid development faults by misconceptions, in particular through reviews. Similarly, Tell et al. (2021), found that design reviews are a key ingredient among practitioners in most hybrid approaches.

Previous work on hybrid methods place a lot of emphasis on back-log management. e.g., a generic hybrid model would have backlog management and three of the following four methods: code review, coding standards, refactoring, and release planning; whereas a “water-scrum-fall” method would involve prototyping, and iteration/sprint review as well as two or the following three methods: code review, coding standards and release planning (Tell et al., 2021). A notable difference is that our candidates do not include backlog management since our candidates have an origin in traditional development processes. There are also similarities in that the previous work calls for code reviews and coding standards, which overlaps with candidates A.4 and A.5 on peer-reviews during development and static code analysis. These two candidates got 23% of the quantitative appraisal by the reference group.

The two run-time candidates that both had a positive qualitative attitude from the focus group, as well as a non-zero investment in the quantitative appraisal, were R.3 and R.4 on monitoring for error detection and prevention, as well as protection against abnormal operating conditions. The focus group would have invested 20% of their budget in these two candidates.

A tool validation test suite (candidate V.4) could be a dynamic part of the stable framework solution, supposed to grow and shrink in coherence with its increments. The suite can ensure that the framework still complies with requirements of the already implemented features when changes or new features are to be introduced by unit testing (V.6) and testing with fault-injection (V.5) can be used. These three candidates were all favored by the reference group, and got a total of 29% of their quantitative appraisal (details in Table 1). The importance of testing is well anchored in previous work on agile and hybrid methods.

Future Work

The findings of this study could be extended in several ways. First, the literature study could be extended to include a larger set of standards and a wider range of publications, to capture industrial perspectives from several different safety-related domains. For example, one future extension is to study how to validate a build tool-chain according to IEC 61508 and other relevant safety-related standards (https://www.iar.com/sv/knowledge/learn/functional-safety/how-to-validate-a-build-toolchain/). For more generalizable results, the focus group could be expanded to capture several different industrial contexts. Further refinement of candidates based on input from a diverse set of groups and industrial contexts would likely increase the general applicability.

Future work could also investigate dynamic validation of the general solution, the candidates, as well as both positive (e.g., reduced amounts of errors or invalid results), and negative (e.g., increased lead-times or reduced innovations) outcomes of the proposed DoDs.

Conclusions

The quality of embedded systems is often demonstrated by test results. Test framework risks are related to masking of problems from detection, erroneous test-system hardware configurations, and omitted feedback on failed tests. These risks may be mitigated with approaches from safety-critial development. However, safety-critical development is often in conflict with agile development. In this case study, we explore how quality assurance for a test framework in an agile non-safety development context could be enhanced by strategies found in safety-critical development. By processing the results of a literature study, candidate solutions to quality assuring the quality assurance tool were identified and divided into four aspects. We also identified the importance of perceiving a test framework, not as a single tool, but as a tool-chain. The interaction sequences through sub-tools can be utilized for analysis and identification of applicable measures. In relation to standards, sub-tools can be classified on an individual basis and confidence argued as the sum of applied measures throughout the tool-chain that is the framework.

A focus group provided insights on implications and perceived industrial value of the proposed candidates. Qualitative data from the focus group identified considerations from an agile industrial perspective: measures for safety do not always entail quality, the level of rigour regarding a tool must be reasonable, effort and gained value must be balanced, and a lowest bar of acceptance—a minimal set of quality assurance activities—should be set. More practical aspects to consider were: the content of reviews and documentation should be clearly defined, the tool environmental conditions should be ensured, it should be possible to distinguish between errors in testware from errors in software, and errors in one test case should not affect subsequent tests or suites. Candidates considered as high value were often related to historical events, for example beneficial candidate solutions included “peer-reviews”, “measures to avoid faults originating as a result of misconceptions”, “performing formal risk and impact analysis”, “developing a monitoring system for error detection and prevention”, “perform tests based on fault injection” and “perform unit tests”. The rejected candidates were perceived as having high effort without apparent gain. Examples of such candidates are “re-develop the entire test framework” and “utilize a safety standard to validate the tool and related processes”. The unified interpretation of qualitative and quantitative results gives a clear indication of what aspects were considered the most important, and where initial efforts should be placed.

Supplemental Information

Supplemental Information 1 Describes a proposed case-specific application of the results

Here we suggest a possible application of the results as an augmented agile process with added rigour through mini V-models controlled by case-specific Definitions of Done.

Click here for additional data file.

Supplemental Information 2 COREQ checklist regarding the conducted focus group

This document answers the checklist described in Tong, A., Sainsbury, P., & Craig, J. (2007). Consolidated criteria for reporting qualitative research (COREQ): a 32-item checklist for interviews and focus groups. International journal for quality in health care, 19(6), 349-357.

The checklist is the substitute for the raw data which cannot be published.

Click here for additional data file.

The work is based on the Master’s thesis (Thörn, 2020) of the first author, where the second and third authors were his supervisors and the fourth author was his examiner.

Additional Information and Declarations

Competing Interests

Author Contributions

Data Availability

1 A test framework is in this case a software development tool for automated software testing. This contains testware with software, documentation, test cases, test data and test environments, which may include physical test-systems that run the software under test (ISTQB, 2016; Strandberg, 2021).

2 The concept of functional safety relates to absence of unacceptable risks and protection against human errors, hardware failures and environmental factors. It involves the identification of possible failures and assigning a tolerance to those (Smith & Simpson, 2004).

3 DoD is an agile concept, a set of criteria to define if a deliverable is done (Silva et al., 2017).

4 A linter is a static code analysis tool that detects suspicious constructions, e.g., incorrect assignments, out of bounds indexing, and dangerous data type combinations Jones (2018).

Jonathan Thörn and Per Erik Strandberg are employed by Westermo Network Technologies. Wasif Afzal is an Academic Editor for PeerJ Computer Science.

Jonathan Thörn conceived and designed the case study, performed the case study, analyzed the data, performed the computation work, prepared figures and/or tables, authored or reviewed drafts of the article, and approved the final draft.

Per Erik Strandberg conceived and designed the case study, performed the case study, prepared figures and/or tables, authored or reviewed drafts of the article, and approved the final draft.

Daniel Sundmark conceived and designed the case study, performed the case study, authored or reviewed drafts of the article, and approved the final draft.

Wasif Afzal conceived and designed the case study, authored or reviewed drafts of the article, and approved the final draft.

The following information was supplied regarding data availability:

The COREQ checklist is available in the Supplemental File in lieu of raw data which could not be made public as the participants’ consent was not obtained at the time it was collected.

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
