# Peer review of "Quality assuring the quality assurance tool: applying safety-critical concepts to test framework development"

_PeerJ Computer Science, doi:10.7717/peerj-cs.1131_

## Round 0.1 · original submission · Major Revisions

The reviewers and I found this paper to be interesting with an important new angle on the topic. Yet, we also see several critical issues that I would suggest you to address before the paper can be accepted. You will find detailed points in the reviews. I want to especially emphasise two improvements:
1) Add a more detailed literature analysis and anchor the whole text better in the existing literature.
2) Improve and detail the focus group in terms of the empirical design, analysis and results.

Reviewer 1 ·

Basic reporting

The language in the paper is clear, unambiguous, and professional, coupled to an acceptable paper structure, and well visualized figures and tables. Sufficient references are provided, but the focus of the paper and associated gaps are not clearly stated (neither in the introduction, nor later). Raw data is not shared, but the methodology is qualitative (meaning that this is not necessarily possible due to privacy issues, or even valuable). If the data was aggregated in several steps these could be valuable to share, but is unlikely it was in this study.

Further details are provided in “General Comments”.

Experimental design

The paper is a “case study”, which could be problematic according to the journal scope description. However, with this the authors mean that they are using a “case study methodology”, i.e. not that they are simply writing up a description of a case. The authors still need to describe the method, and motivate the case, better, but I believe it could still be acceptable (if I interpret the scope of the journal correctly). The research questions are well defined, but (as noted previously) should be better motivated in regard to current research gaps. Otherwise, it is difficult to assess whether the research is relevant and meaningful. The methodology section and supplementary material describe the focus group approach in the case study sufficiently. The approach is not strong considering what is described in the supplementary material, but described to a sufficient level to be possible to replicate.

Further details are provided in “General Comments”.

Validity of the findings

The described “literature review” has low validity, and the associated results are overreported. They would be better reported as a simple table, stating what is essentially a focus group script (there is no credible motivation for dedicating a whole section to this). The paper also contains a focus group study, which is possible to replicate. According to the supplementary material the approach is weak in regard to sample choice (convenience, but no strong argument for case), sample size (small), not iterating, not recording audio/video from the focus group, data saturation not being ensured, no checking of the results by the participants, and an unclear approach to coding and thematic analysis. Subsequent to the results a set of proposed guidelines are put forth. However, they have no strong relationship to the empirical evidence from the investigation, and should be reduced to avoid overstating the results.

Further details are provided in “General Comments”.

Additional comments

A) Minor: The authors state that: “Standards for functional safety often rely on a plan-driven process with predefined phases.”
This is a bit of an overstatement, which the authors aggravate by stating in Section 2: “… where development is a strictly sequential process through predefined phases (Jonsson et al., 2012).”
This is not necessarily true, particularly the part regarding “strictly sequential”. (It might still be a planned, heavy process.) I suggest you decrease the strength of the statement through the caveat “… even if these are not necessarily meant to be conducted in a strictly sequential fashion” (or similar), as well as remove/rephrase further statement on this being “strict”.

B) Major: The authors state that: “Available research on combining agile and plan-driven methods is mostly from the perspective of utilizing agile practices in existing plan-driven processes. There is a research gap with respect to implementing plan-driven practices in agile processes to increase confidence and quality, the goal of this paper is to fill parts of that gap.”
Please include relevant references, be more specific on what parts of the gap you are aiming to close and clarify why this is important. The above is a very broad statement, and both your abstract and the second paragraph of the introduction suggest you are aiming for a narrow contribution. Essentially, a reader will want to have an argument for why “strategies for increased confidence and quality in tools used for automated software testing in non-safety development may be found or created from concepts and strategies related to safety-critical development”. I agree this is probably true, but you do not provide an argument for it.
Presumably, such an argument could be built based on what “typical” test activities/tooling used in industry for non-safety critical software do not provide.

C) Minor: The authors state that: “Popular among safety standards is to describe this sequential flow through the V-model (Asplund, 2014), illustrated in Figure 1.”
This is also a bit of an overstatement, as many standards try to be process agnostic. I suggest you decrease the strength of the statement by referring to these standards as instead being influenced by the thinking behind the V-Model (or similar).

D) Minor: In regard to the paragraph starting with: “Safety Integrity Levels (SILs)…”
I suggest that you add the caveat that the definitions of contents of these levels can be quite different among the standards. (Aerospace is e.g., very different from automotive in this regard.) Although you mention this in Subsection 2.2 in regard to the tools, a reader will probably start to wonder already here.

E) Major: A better reference on case studies by Runeson is his book from 2012. I suggest you take a look at Table 3.1 in Chapter 3 in that book and make sure you are covering the methodology thoroughly in your paper.

F) Major: The authors state: “The studied case in the research is defined as the industrial partner and the products developed. The unit of analysis is defined as the development and maintenance of the test framework, utilized at the industrial partner for the execution of manual and automated tests of produced products.”
However, there is no discussion of whether this is a suitable case and/or unit of analysis. Why should a reader trust the results from the case, and why would not a different case give completely different result? You need to provide relevant and convincing arguments in regard to this.

G) Major: You are not describing or providing enough evidence to suggest you performed a systematic literature review in Subsection 4.1. It is thus impossible to replicate what you have done in regard to reviewing literature, and the scientific value of this is thus low (beyond serving as a background and related work). You should remove this section, or provide more information/evidence to convince a reader it was likely to arrive at comprehensive results.

H) Major: Following up on (H), as the literature review is not systematic, the whole of Section 5 is basically a description of your focus group script. You can put if forth as a result if you want (perhaps as a table referred to from Section 4). However, as the literature review is not systematic you should be very careful not to overstate the results. I suggest you move what you need to Section 2 or Section 3, and put the rest of the text in an appendix.

I) Major: There is no strong foundation for suggesting the guidelines put forth in Section 7. It is unclear why you would want to suggest these guidelines, as you are not testing them. It would thus make more sense to have these guidelines in the start of a subsequent paper which validates them.
As Section 8 is mostly repeating the results, I suggest you remove Section 7 and add a convincing discussion to support the parts of the guidelines you can motivate separately in what is now Section 8. (In other words, be specific and discuss each result instead of proposing a large concept that is only vaguely related to the each result.) If you want to keep the proposed guidelines, then you need to be much clearer on how you arrive to the proposed guidelines (and only the proposed guidelines) based on the results. Currently this is very vague.

Reviewer 2 ·

Basic reporting

The topic and approach angle is interesting!
There could be more literature references as e.g.
- Joint System safety engineering handbook
- Functional safety and proof of compliance
Topics in these documents that are relevant are:
culture, Culture, tool process, updated tool requirements IEC 61508, and Definition of ready/done.

I also would like to know more regarding relevant analysis that should be performed to ensure relevant tests, automatic tests, tool chain approach

Experimental design

The open questions in the Focus group did not result in a clear answer. And as a result of these comments, the Conclusion was not as concrete as I hoped.

Validity of the findings

See comments on section 1 of this review above. Could do more literature study

Definition of done is strongly linked to the definition of ready (DoR). DoR is so far not mentioned.

Additional comments

The conclusion could be more specific on the pitfalls, disadvantage and benefits.

---

## Round 0.2 · Minor Revisions

Please focus on discussing the weaknesses in the methodology. Reviewer 1 gave you good and detailed hints to improve this.

Reviewer 1 ·

Basic reporting

The paper is still pleasant to read. The improved narrative and structure of the paper, and the move/removal of overstated results, further helps the reader focus and appreciate the contributions of the paper. The related, previous comments by me regarding the basic reporting have been acceptably and honestly answered/dismissed through a reasonable effort by the authors.

Experimental design

The authors now position their study in regard to other existing strands of research, implying that their contribution is novel and valuable. Furthermore, the description of the methodology (case study) is now adequately structured.

Validity of the findings

However, the paper still does not adequately argue in regard to the weaknesses apparent in the methodology. While most studies are far from perfect, it is important that the authors clarify the weaknesses of their study enough to allow readers to (by themselves) decide how much the conclusions can be trusted. That said, I think the changes required to reach and adequate state is relatively small.

Further details are provided in “Additional Comments”.

Additional comments

A) For Your Information: I wonder whether you need to include this statement:
“Applying constraints on the development process was seen as less compatible with agile development, which is why this article focuses on standards in the first group.”
While I understand the comment and appreciate your wish to be transparent, it might be a bit confusing to the average reader (not reading the paper in detail).

B) Major: The paper now clearly states the type of case used, but it does not argue why this case is suitable:
“A second rationale for a single case is when the case is an extreme or a unique case, however, this does not apply in our context, rather our case is representative or typical (third rationale). The industrial partner and the products developed are typical of many embedded system manufacturers, both in the same domain of industrial communication equipment and in other similar domains. Thus, the results taken from this case are assumed to be typical of the experiences of an average embedded system manufacturer.”
While you can of course do a lot to present a strong positive argument, it should be sufficient/easier to simply provide a negative one here. In other words, you can, in Section 6.3, discuss the most obvious conclusions which could be different/invalid/less strong in more extreme situations. Purely as an example, embedded software developers often end up being the only expert in “their” system components or support tools/toolchains. Presumably this would decrease the value of reviews, or at least imply they should be focused on particular problems. (I could offer a few references on this and other possible talking points, but I think you have a few to choose from already.) If you provide such a small discussion, it should be enough to give a reader a feeling for the most important limits of your conclusions.

C) Major: There are rudimentary responses in your text in regard to the validity concerns I raised, but they are not very clear. Please clarify:
C.1) “…sample choice (convenience, but no strong argument for case)” -> See Comment (B), are you missing input from some important stakeholders due to who you picked for the group? (Note that the triangulation that you make use of and e.g., discuss in Section 6.3 does not help in regard to omissions in regard to sample choice.)
C.2) “sample size (small)” -> See Comment (B), are you missing input from some important stakeholders due to how many you interviewed? (Note that the triangulation that you make use of and e.g., discuss in Section 6.3 does not help in regard to omissions in regard to sample size.)
C.3) “not iterating” and “data saturation not being ensured” -> The triangulation is a possible argument here. However, while you refer to checking that there were no more input in the research process, it is unclear if this was the case. E.g. “The focus group ended with a summarizing event, asking the participants if there were any candidates they had expected to be presented but that were missing, and if they could share any other thoughts or ideas regarding the material that had been presented to them.” (Did the participants state there was nothing more to say?)
C.4) “not recording audio/video from the focus group” -> You do not have to record audio/video, and you were not able to in this case. Sure. However, even Krueger states concerns when using notes and memory (it not being suitable for novices, etc.). If you handled these concerns, then state so.
I still think your analysis of the notes are a bit unclear, but if you were indeed doing this together I guess the risk you missed anything should be small.

D) Minor: The mentions of DoDs in Section 6.3 and 6.5 does not connect well with the rest of the text, now that you have moved much of the discussion of DoDs to an Appendix. I suggest you rephrase/reduce these parts.

---

## Author Rebuttal · Round 0.2

**Response Letter to the manuscript:**
**"Quality assuring the quality assurance tool: A case study on applying safety-critical concepts to test framework development"**

Dear Editor and Reviewers,

We are grateful to you for the constructive comments received on our manuscript. In this letter, we provide our responses and the changes done to address the given comments. Please note that wherever applicable, we have merged similar comments to avoid repetition.

We hope that these changes have improved the quality of the manuscript and that it is now in publishable form.

Sincerely,
Authors
* * *
**Editor's Comments**
* * *
**Editor comment 1 and 2**

1. Add a more detailed literature analysis and anchor the whole text better in the existing literature.

2. Improve and detail the focus group in terms of the empirical design, analysis and results.

*Authors' response and action:*

Thanks for the two summary comments; now after addressing both reviewers' comments, we have addressed these two issues. In particular, we made major changes in the description of the method (Section 4), and de-emphasized the significance of the literature study part. We also, more thoroughly, discussed the quality and limitations of the focus group, and made many other minor changes, e.g. removed over-statements of the suggested case-specific approach by moving it to the Appendix. Please refer to our responses to reviewers' comments below for further details.
* * *
**Reviewer # 1's Comments**
* * *
**Reviewer 1 comment 1**

1. Sufficient references are provided, but the focus of the paper and associated gaps are not clearly stated (neither in the introduction, nor later).
The research questions are well defined, but (as noted previously) should be better motivated in regard to current research gaps. Otherwise, it is difficult to assess whether the research is relevant and meaningful.

*Authors' response and action:*
We have now sharpened the focus and the identified gap that we aim to fill. This is now stated clearly in two sections: Section 1, 3rd and 4th paragraphs and Section 3.2.

**Reviewer 1 comment 2**

2. Raw data is not shared, but the methodology is qualitative (meaning that this is not necessarily possible due to privacy issues, or even valuable). If the data was aggregated in several steps these could be valuable to share, but is unlikely it was in this study.

*Authors' response and action:*
The raw data could not be shared due to two reasons: (1) no permission being given from the company involved in this work due to confidentiality and (2) recording of focus group feedback in Swedish language, thus requiring extensive English translation. This limitation was communicated to PeerJ during submission and a work-around was reached where we provided answers to the COREQ checklist (https://pubmed.ncbi.nlm.nih.gov/17872937/).
The reviewer is right that the data was not aggregated in several steps.

**Reviewer 1 comment 3**

3. The paper is a "case study", which could be problematic according to the journal scope description. However, with this the authors mean that they are using a "case study methodology", i.e. not that they are simply writing up a description of a case.

*Authors' response and action:*
This is correct that we are not simply describing the case but actually following the guidelines of reporting case studies in software engineering from start to end, including analysis of gathered data to answer the posed research questions.

**Reviewer 1 comment 4**

4. The authors still need to describe the method, and motivate the case, better, but I believe it could still be acceptable (if I interpret the scope of the journal correctly).

*Authors' response and action:*
The method section is now thoroughly revised to follow recommended guidelines, please refer to Section 4 and all the accompanying subsections.

We also would like to comment that, as far as we have understood Peerj Computer Science scope of topics, it does not restrict submissions based on applied research methods. We

also believe that a case study is quite a typical research method applied in software engineering (software engineering is a recognized subject in PeerJ Computer Science).

**Reviewer 1 comment 5**

5.  The described "literature review" has low validity, and the associated results are overreported. They would be better reported as a simple table, stating what is essentially a focus group script (there is no credible motivation for dedicating a whole section to this).

G) Major: You are not describing or providing enough evidence to suggest you performed a systematic literature review in Subsection 4.1. It is thus impossible to replicate what you have done in regard to reviewing literature, and the scientific value of this is thus low (beyond serving as a background and related work). You should remove this section, or provide more information/evidence to convince a reader it was likely to arrive at comprehensive results.

H) Major: Following up on (H), as the literature review is not systematic, the whole of Section 5 is basically a description of your focus group script. You can put it forth as a result if you want (perhaps as a table referred to from Section 4). However, as the literature review is not systematic you should be very careful not to overstate the results. I suggest you move what you need to Section 2 or Section 3, and put the rest of the text in an appendix.

*Authors' response and action:*

We have now re-structured the manuscript sections to cater for the suggestion of removing the literature review section. In the old submission, the literature study results were presented in Section 5. Now, there is no dedicated section of literature review, rather we have two subsections on it in Section 3. These subsections are "Section 3.3: Approaches for Increased Tool Confidence" which introduces the 48 approaches and how they were derived from their respective references, and "Section 3.4: Summary of Approaches from Previous Work", which summarise how the 48 approaches became 22 candidate solutions. The results are also reported in Table 1. The old "Section 5.4: The Test Tool as a Tool-Chain" is now moved as "Section 5.1.1 The Test Tool as a Tool-Chain" under "Section 5.1 Focus Group Preparation" since the tool-chain concept acted as a preparatory step in conducting the focus group.

**Reviewer 1 comment 6**

6.  The paper also contains a focus group study, which is possible to replicate. According to the supplementary material the approach is weak in regard to sample choice (convenience, but no strong argument for case), sample size (small), not iterating, not recording audio/video from the focus group, data saturation not being ensured, no checking of the results by the participants, and an unclear approach to coding and thematic analysis.

*Authors' response and action:*

Some of these limitations and our actions to overcome them are discussed in Section 6.3. As we write there, the focus group also had a quantitative part (described in Section 4.8.2), where the qualitative feedback got validated and acted as a proxy for saturation. In Section 4.8.2, we also write how gathered data was analysed. We also mention that recording audio/video is not a necessity according to one reference (Krueger (2014)) and regardless,

we did not have the necessary permission from the company to record. In addition, "Section 4.9 Quality Assurance, Validity and Reliability", we further list measures to enhance the quality, validity and reliability of case study design and focus group.

**Reviewer 1 comment 7**

7. Subsequent to the results a set of proposed guidelines are put forth. However, they have no strong relationship to the empirical evidence from the investigation, and should be reduced to avoid overstating the results.
   Major: There is no strong foundation for suggesting the guidelines put forth in Section 7. It is unclear why you would want to suggest these guidelines, as you are not testing them. It would thus make more sense to have these guidelines in the start of a subsequent paper which validates them.
   As Section 8 is mostly repeating the results, I suggest you remove Section 7 and add a convincing discussion to support the parts of the guidelines you can motivate separately in what is now Section 8. (In other words, be specific and discuss each result instead of proposing a large concept that is only vaguely related to each result.) If you want to keep the proposed guidelines, then you need to be much clearer on how you arrive at the proposed guidelines (and only the proposed guidelines) based on the results. Currently this is very vague.

*Authors' response and action:*
We agree to the suggestion of removing the section on proposed guidelines. In the discussion section (now Section 6), we have a new "Section 6.4: Revisiting Related Work from the Perspective of the Focus Group" which discusses the case-specific results. Also, in Section 6.2, last paragraph, we discuss an augmented agile process. We also have an extended Appendix A with added text on practical application of definition of done using the mini-v model, previously introduced as part of the proposed guidelines.

**Reviewer 1 comment 8**

8. A) Minor: The authors state that: "Standards for functional safety often rely on a plan-driven process with predefined phases."
   This is a bit of an overstatement, which the authors aggravate by stating in Section 2: "… where development is a strictly sequential process through predefined phases (Jonsson et al., 2012)."
   This is not necessarily true, particularly the part regarding "strictly sequential". (It might still be a planned, heavy process.) I suggest you decrease the strength of the statement through the caveat "… even if these are not necessarily meant to be conducted in a strictly sequential fashion" (or similar), as well as remove/rephrase further statements on this being "strict".

*Authors' response and action:*
We agree. We have now reworked the sentences to decrease the strength of the arguments. Please refer to Section 1, first sentence of the third paragraph and Section 2, first paragraph, last two sentences.

**Reviewer 1 comment 9**

9. B) Major: The authors state that: "Available research on combining agile and plan-driven methods is mostly from the perspective of utilizing agile practices in

existing plan-driven processes. There is a research gap with respect to implementing plan-driven practices in agile processes to increase confidence and quality, the goal of this paper is to fill parts of that gap."

Please include relevant references, be more specific on what parts of the gap you are aiming to close and clarify why this is important. The above is a very broad statement, and both your abstract and the second paragraph of the introduction suggest you are aiming for a narrow contribution. Essentially, a reader will want to have an argument for why "strategies for increased confidence and quality in tools used for automated software testing in non-safety development may be found or created from concepts and strategies related to safety-critical development". I agree this is probably true, but you do not provide an argument for it.

Presumably, such an argument could be built based on what "typical" test activities/tooling used in industry for non-safety critical software do not provide.

*Authors' response and action:*
"Section 1: Introduction" has been reworked to make the focus of the paper clear and specific. We also mention the gap clearly and refer the reader to a dedicated Section 3.2 where we dedicate the entire section to position our work in light of previous work. Please refer to paragraphs 3 and 4 in Section 1.

**Reviewer 1 comment 10**

10. C) Minor: The authors state that: "Popular among safety standards is to describe this sequential flow through the V-model (Asplund, 2014), illustrated in Figure 1."
This is also a bit of an overstatement, as many standards try to be process agnostic. I suggest you decrease the strength of the statement by referring to these standards as instead being influenced by the thinking behind the V-Model (or similar).

*Authors' response and action:*
We agree. We have reworked the sentence to reduce the strength of the statement. Please refer to Section 2, first paragraph, last sentence.

**Reviewer 1 comment 11**

11. D) Minor: In regard to the paragraph starting with: "Safety Integrity Levels (SILs)…"
I suggest that you add the caveat that the definitions of contents of these levels can be quite different among the standards. (Aerospace is e.g., very different from automotive in this regard.) Although you mention this in Subsection 2.2 in regard to the tools, a reader will probably start to wonder already here.

*Authors' response and action:*
We agree. Now we have reworked a couple of sentences. Please refer to Section 2.1, third paragraph, first three sentences.

**Reviewer 1 comment 12**

12. E) Major: A better reference on case studies by Runeson is his book from 2012. I suggest you take a look at Table 3.1 in Chapter 3 in that book and make sure you are covering the methodology thoroughly in your paper.

*Authors' response and action:*
Please refer to "Section 4: Research Methodology", which is now reworked to include information/headings suggested in Table 3.1 of Runeson's book of 2012.

**Reviewer 1 comment 13**

13. F) Major: The authors state: "The studied case in the research is defined as the industrial partner and the products developed. The unit of analysis is defined as the development and maintenance of the test framework, utilized at the industrial partner for the execution of manual and automated tests of produced products."

    However, there is no discussion of whether this is a suitable case and/or unit of analysis. Why should a reader trust the results from the case, and why would not a different case give completely different result? You need to provide relevant and convincing arguments in regard to this.

*Authors' response and action:*
Please refer to "Section 4.3: Case Selection Strategy", which now motivates our case selection strategy in light of *rationale for single-case designs* from (Yin, 2009).
* * *
**Reviewer # 2's Comments**
* * *
**Reviewer 2 comment 1**

1. There could be more literature references as e.g.
   - Joint System safety engineering handbook
   - Functional safety and proof of compliance
   Topics in these documents that are relevant are:
   culture, Culture, tool process, updated tool requirements IEC 61508, and Definition of ready/done.

   See comments on section 1 of this review above. Could do more literature study

   Definition of done is strongly linked to the definition of ready (DoR). DoR is so far not mentioned.

*Authors' response and action:*
We thank the reviewer for suggesting further topics relevant to the paper focus. Out of these topics, we now highlight "definition of ready" as being related to the "definition of done" as a footnote on page 25. We further highlight in "Section 6.5 Future Work" that an interesting extension to our work is to validate a tool-chain according to IEC61508 and other relevant safety standards. We further acknowledge in Section 6.5 that a more thorough literature study will capture a wide range of publications and standards.

**Reviewer 2 comment 2**

2. I also would like to know more regarding relevant analysis that should be performed to ensure relevant tests, automatic tests, tool chain approach.

*Authors' response and action:*
The candidate solutions from the literature study and the focus group are summarised in Table 1. These are grouped into four main aspects; the most relevant ones for this comment are "Analysis" and "Verification & Validation". Each of these candidate solutions have explanations based on the approaches in "Section 5.2 Qualitative Results of Focus Groups". For example, one of the candidate solutions from "*Verification & Validation*" is "*Formally prove that tool outputs conforms to specification*", which came out of the discussion on what is the performance of the regression testing tool applied in the company. Similarly, another candidate solution from the same main aspect is "*Perform unit tests on all modules and tools in tool-chain(s)*" that points to a relevant analysis that needs to be done for achieving the solution.

**Reviewer 2 comment 3**
3. The open questions in the Focus group did not result in a clear answer. And as a result of these comments, the Conclusion was not as concrete as I hoped.

   The conclusion could be more specific on the pitfalls, disadvantages and benefits.

*Authors' response and action:*
We partially agree that the focus group did not result in a clear answer. Partly, the candidate solutions are broad guidelines (for example, "*Apply measures to avoid development faults introduced by misconceptions*"), but partly they are prescriptive too (for example, "*Formally prove that tool outputs conforms to specification*"). The focus group had both qualitative and quantitative phases (Sections 5.2 and 5.3 respectively). Further, the candidate solutions are summarised, both in Table 1 and in "Section 5.5 Summary of Results and Final Candidates". The approaches on which the candidate solutions are dependent are explained in "Section 5.2 Qualitative Results of Focus Groups" to further clarify the results.

The conclusions now include examples of candidate solutions that were found to be of high value (beneficial) as well as those that were down rated.
* * *
**End of Response Letter**

---

## Round 0.3 · accepted · Accept

Thank you for addressing the remaining comments from the reviewer. We are now all happy with the article, and in particular the improved presentation. The article is now ready for publication!

Reviewer 1 ·

Basic reporting

See previous review comments.

I think the authors have responded well to the previous review comments. Readers can now more easily evaluate the strength of the paper by themselves.

Experimental design

See previous review comments.

Validity of the findings

See previous review comments.

Additional comments

No additional comments.

---

## Author Rebuttal · Round 0.3

# Second Response Letter to the manuscript:
## "Quality assuring the quality assurance tool: A case study on applying safety-critical concepts to test framework development"

Dear Editor and Reviewer,
We thank you for this second round of professional and constructive review comments. In this letter, we provide our responses and the changes done to address the given comments.

We hope that these changes have improved the quality of the manuscript and that it is now in publishable form.

Sincerely,
Authors
September 2022
* * *
**Reviewer # 1's Comments**
* * *
**Basic reporting, and Experimental design.**
No change required.

**Validity of the findings**

- However, the paper still does not adequately argue in regard to the weaknesses apparent in the methodology. While most studies are far from perfect, it is important that the authors clarify the weaknesses of their study enough to allow readers to (by themselves) decide how much the conclusions can be trusted. That said, I think the changes required to reach and adequate state is relatively small.

*Authors' response and action:*
We thank the reviewer for this valuable comment. We have made several changes to the paper, as detailed in the answers below.

**Additional Comments A**

- For Your Information: I wonder whether you need to include this statement: "Applying constraints on the development process was seen as less compatible with agile development, which is why this article focuses on standards in the first group." While I understand the comment and appreciate your wish to be transparent, it might be a bit confusing to the average reader (not reading the paper in detail).

*Authors' response and action:*
As suggested, we removed the statement in section 2.1.

**Additional Comments B**

- The paper now clearly states the type of case used, but it does not argue why this case is suitable:
  "A second rationale for a single case is when the case is an extreme or a unique case, however, this does not apply in our context, rather our case is representative or typical (third rationale). The industrial partner and the products developed are typical of many embedded system manufacturers, both in the same domain of industrial communication equipment and in other similar domains. Thus, the results taken from this case are assumed to be typical of the experiences of an average embedded system manufacturer." While you can of course do a lot to present a strong positive argument, it should be sufficient/easier to simply provide a negative one here. In other words, you can, in Section 6.3, discuss the most obvious conclusions which could be different/invalid/less strong in more extreme situations. Purely as an example, embedded software developers often end up being the only expert in "their" system components or support tools/toolchains. Presumably this would decrease the value of reviews, or at least imply they should be focused on particular problems. (I could offer a few references on this and other possible talking points, but I think you have a few to choose from already.) If you provide such a small discussion, it should be enough to give a reader a feeling for the most important limits of your conclusions

*Authors' response and action:*
We would like to argue that the suitability of the case is typical of embedded system manufacturers. However, to acknowledge this potential limitation, we added a paragraph on generalizability in Section 6.3, including an example where one of our conclusions might not be applicable. We also added two references (second-last paragraph in Section 6.3) to back the value of industrial case studies, despite limitations on generalizability.

**Additional Comments C.1 and C.2**

- "…sample choice (convenience, but no strong argument for case)" -> See Comment (B), are you missing input from some important stakeholders due to who you picked for the group? (Note that the triangulation that you make use of and e.g., discuss in Section 6.3 does not help in regard to omissions in regard to sample choice.)
- "sample size (small)" -> See Comment (B), are you missing input from some important stakeholders due to how many you interviewed? (Note that the triangulation that you make use of and e.g., discuss in Section 6.3 does not help in regard to omissions in regard to sample size.)

*Authors' response and action:*
We agree that the sample might be perceived as limited, however, it was representative of the roles involved in developing WeOS. We updated the 2'nd paragraph of Section 6.3 with the following:

> Involving a larger number of participants would most likely have given a greater sample size and more diverse feedback. But in terms of diversity of roles participating in the focus group, only a few relevant ones were omitted, e.g. no

software architects were present but they typically have similar competences as senior developers.

**Additional Comments C.3**

- "not iterating" and "data saturation not being ensured" -> The triangulation is a possible argument here. However, while you refer to checking that there were no more input in the research process, it is unclear if this was the case. E.g. "The focus group ended with a summarizing event, asking the participants if there were any candidates they had expected to be presented but that were missing, and if they could share any other thoughts or ideas regarding the material that had been presented to them." (Did the participants state there was nothing more to say?)

*Authors' response and action:*
We clarified this by adding a new last sentence in 4.9, by extending and clarifying 5.4, and by moving a sentence from 5.3 to 5.4.

**Additional Comments C.4**

- "not recording audio/video from the focus group" -> You do not have to record audio/video, and you were not able to in this case. Sure. However, even Krueger states concerns when using notes and memory (it not being suitable for novices, etc.). If you handled these concerns, then state so.
  I still think your analysis of the notes are a bit unclear, but if you were indeed doing this together I guess the risk you missed anything should be small.

*Authors' response and action:*
We have simplified the writeup of the *data collection* in Section 4.8.2, stating that we relied on multiple persons taking the notes to counter downsides of note-based data collection.

**Additional Comments D**

- The mentions of DoDs in Section 6.3 and 6.5 does not connect well with the rest of the text, now that you have moved much of the discussion of DoDs to an Appendix. I suggest you rephrase/reduce these parts.

*Authors' response and action:*
We agree and have reduced the last paragraph of section 6.3 and the last paragraph of section 6.5.